# Equivariant Graph Mechanics Networks with Constraints

**Wenbing Huang**[*1], **Jiaqi Han**[*2][†] **Yu Rong**[⊠3], **Tingyang Xu**[3], **Fuchun Sun**[⊠2], **Junzhou Huang**[4]

[1] Institute for AI Industry Research (AIR), Tsinghua University

[2] Beijing National Research Center for Information Science and Technology (BNRist),
Department of Computer Science and Technology, Tsinghua University

[3] Tencent AI Lab

[4] Department of Computer Science and Engineering, University of Texas at Arlington

`hwenbing@126.com, hanjq21@mails.tsinghua.edu.cn, yu.rong@hotmail.com`
`tingyangxu@tencent.com, fcsun@mail.tsinghua.edu.cn, jzhuang@uta.edu`

## Abstract

Learning to reason about relations and dynamics over multiple interacting objects is a challenging topic in machine learning. The challenges mainly stem from that the interacting systems are exponentially-compositional, symmetrical, and commonly geometrically-constrained. Current methods, particularly the ones based on equivariant Graph Neural Networks (GNNs), have targeted on the first two challenges but remain immature for constrained systems. In this paper, we propose Graph Mechanics Network (GMN) which is combinatorially efficient, equivariant and constraint-aware. The core of GMN is that it represents, by generalized coordinates, the forward kinematics information (positions and velocities) of a structural object. In this manner, the geometrical constraints are implicitly and naturally encoded in the forward kinematics. Moreover, to allow equivariant message passing in GMN, we have developed a general form of orthogonality-equivariant functions, given that the dynamics of constrained systems are more complicated than the unconstrained counterparts. Theoretically, the proposed equivariant formulation is proved to be universally expressive under certain conditions. Extensive experiments support the advantages of GMN compared to the state-of-the-art GNNs in terms of prediction accuracy, constraint satisfaction and data efficiency on the simulated systems consisting of particles, sticks and hinges, as well as two real-world datasets for molecular dynamics prediction and human motion capture.

## 1 Introduction

Representing and reasoning about the relations and dynamics of a group of interacting objects is among the core aspects of human intelligence (Tenenbaum et al., 2011; Ding et al., 2021; Gan et al., 2021). As a motivating example, we consider the N-body system (Kipf et al., 2018) where the movement of a single charged particle is attracted or repelled by other charged particles. Physicists have revealed that this process can be modeled by Newton's laws along with the Coulomb force. One may wonder, however,

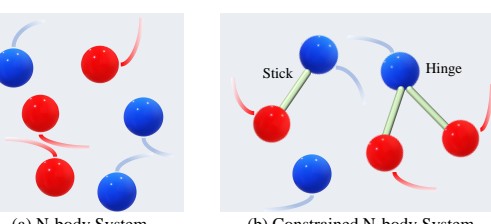

(a) N-body System     (b) Constrained N-body System

Figure 1: N-body vs. constrained N-body (red/blue balls denote positive/negative charges).

if we can teach a machine to rediscover the underlying physics by solely observing the particles' states. This thinking has inspired the study of learning to model interacting systems, which now is a prevailing topic in machine learning (Battaglia et al., 2016; Thomas et al., 2018; Köhler et al., 2019; Sanchez-Gonzalez et al., 2019; Fuchs et al., 2020; Martinkus et al., 2021; Satorras et al., 2021).

---

[*]Equal contributions: Wenbing Huang and Jiaqi Han; ⊠ Corresponding authors: Yu Rong and Fuchun Sun.
[†]This work is done when Jiaqi Han works as an intern in Tencent AI Lab.

Learning to model interacting systems is challenging. First, the systems are combinatorially complex, on account of that objects can be composed in combinatorially many possible arrangements (Battaglia et al., 2016). This challenge, to some extent, can be addressed by making use of Graph Neural Networks (GNNs) (Wu et al., 2020). By regarding objects as nodes and interactions as edges, GNNs extract information via message passing, which is able to characterize arbitrarily ordered objects and combinatorial relations. The second challenge is related to an important symmetry in physics: the model we use should be equivariant to any Euclidean transformation (translation/reflection/rotation) of the input. This complies with the fact that physics rules keep unchanged regardless of the reference coordinate system. Several works (Fuchs et al., 2020; Satorras et al., 2021) have investigated equivariance upon GNNs and exhibited remarkable benefits on N-body.

Another challenge, though less explored, is that the systems could be geometrically constrained. Geometric constraints arise in common practical systems, for example, in robotics where the joints of mechanical arms should be linked one by one, or in biochemistry where the atoms of molecules are connected by chemical bonds. When modeling these constrained systems, it is crucial to enforce the model to output legal predictions. For instance, the lengths/angles of chemical bonds in a molecular closely determine its chemical property which will be changed dramatically if the structural constraints are broken. As mentioned above, equivariant GNNs (Fuchs et al., 2020; Satorras et al., 2021) have achieved desirable performance on the N-body system—this system, nevertheless, lacks of constraint. Considering constraints is not easy, as the dynamics of all elements within a constrained system evolve in a joint and complex manner. Unfortunately, to the best of our knowledge, there is no research (particularly among equivariant GNN methods) that learns to model the dynamical systems of multiple interacting objects under geometrical constraints.

In this paper, we propose Graph Mechanics Network (GMN) that can tackle the above three challenges simultaneously: **I.** To cope with the combinatorial complexity, GMN takes advantage of graph models to encode the states of and interactions between objects, analogous to previous approaches. **II.** For the constraint satisfaction, GMN resorts to generalized coordinates, a well known notion in conventional mechanics to model the dynamics of structural objects. Here, a structural object (such as the stick and hinge in Fig. 1) is defined as a set of multiple rigidly connected particles. In GMN, the constraints are implicitly encoded in the forward kinematics that describes the Cartesian states as the function of generalized coordinates. This strategy can inherently maintain the constraints and requires no external regulation. **III.** GMN is equivariant. In GMN, we need to compute the interaction forces in the Cartesian space (see Eq. 5), and infer the accelerations of the generalized coordinates based on the inverse dynamics (see Eq. 6), both of which are learned by a general form of equivariant functions (see Eq. 11). Notably, the proposed equivariant formulation is more efficient to compute compared to the previous versions based on spherical harmonics (Thomas et al., 2018; Fuchs et al., 2020), or more general than the one developed by EGNN (Satorras et al., 2021). More importantly, we have theoretically discussed when and how our formulation can universally approximate any equivariant function.

To evaluate the advantages of GMN, we have constructed a simulated dataset composed of three types of objects: particles, sticks and hinges, which is a complex case of the N-body system (Kipf et al., 2018) and can be used as building blocks for common systems. Under various scenarios in terms of different ratios of object types and different numbers of training data, we empirically verify the superiority of GMN compared to state-of-the-art models in prediction accuracy, constraint satisfaction and data efficiency. In addition, we also test the effectiveness of GMN on two real-world datasets: MD17 (Chmiela et al., 2017) and CMU Motion Capture (CMU, 2003).

## 2 RELATED WORK

Learning to simulate complex physical systems has been shown to greatly benefit from using GNNs. Interaction Network (IN) proposed by Battaglia et al. (2016) is the first attempt for this purpose, and it can be deemed as a special kind of GNNs to learn how the system interacts and how the states of particles evolve. Later researches have extended IN in different aspects: HRN (Mrowca et al., 2018) utilizes hierarchical graph convolution for tackling objects of various geometrical shapes and materials, NRI (Kipf et al., 2018) further explicitly infers the interactions with the help of a variational auto-encoder, and Hamiltonian graph networks (Sanchez-Gonzalez et al., 2019) equip GNNs with ordinary differential equations and Hamiltonian mechanics for energy conservation. However,

all above approaches have ignored the symmetry in physics and the GNN models they use are not Euclidean equivariant. There is a subset of models (Ummenhofer et al., 2019; Sanchez-Gonzalez et al., 2020; Pfaff et al., 2020) that partially implement symmetries, but they only enforce translation equivariance but not rotation equivariance. On the other hand, it is nontrivial to enforce rotation equivariance. Tensor-Field networks (Thomas et al., 2018) uses filters built from spherical harmonics to allow 3D rotation equivariance, and this idea has been developed by SE(3) Transformer (Fuchs et al., 2020) that further takes the attention mechanism into account. Anther class of works (Finzi et al., 2020a; Hutchinson et al., 2021) resorts to the Lie convolution for the equivariance on any Lie group, based on lifting and sampling. Recently, EGNN (Satorras et al., 2021) has proposed a simple yet effective form of equivariant message passing on graphs, which does not require computationally expensive higher-order representations while still achieving better performance on N-body.

As interpreted before, ensuring geometrical constraints is crucial for many practical systems, which, nevertheless, is seldom investigated in aforementioned works. Although several attempts (Yang et al., 2020; Finzi et al., 2020b) have been proposed for learning to enforce constraints, they explicitly augment the training loss with soft Lagrangian regulation and thus are completely data-driven and have no guarantee of generalization for limited training data. DeLaN (Lutter et al., 2019) also employs generalized coordinates to describe the kinematics of the rigid object. Nevertheless, it only target on the physical process of a single object, other than the complex systems with multiple rigid and structural objects that are the focus of this paper. In DPI-Net (Li et al., 2018), the BoxBath task does share the similar strategy to us by first predicting the canonical coordinates of the box and then using the forward kinematic model to obtain the Cartesian positions. However, the passing messages in DPI-Net are scalars other than directional vectors (positions, velocities, and accelerations) used in our work. After all, both DeLaN and DPI-Net never study equivariant models.

## 3 GRAPH MECHANICS NEURAL NETWORK

We begin with introducing the N-body system (Kipf et al., 2018), as illustrated in Fig. 1 (a). This system consists of $N$ interacting particles $\{P_i\}_{i=1}^N$ of the same mass, and the kinematics states of each particle are defined as $\boldsymbol{S}_i = (\boldsymbol{x}_i, \boldsymbol{v}_i)$, where $\boldsymbol{x}_i, \boldsymbol{v}_i \in \mathbb{R}^3$ are the position and velocity vectors, respectively. There could be certain non-vector information of each particle (such as charge), which is represented by a $c$-channel feature $h_i \in \mathbb{R}^c$ [1]. Suppose the system we study is conservative and the dynamics is driven by the interaction force $\boldsymbol{f}_{ij} \in \mathbb{R}^3$ between any pair of particles $i$ and $j$. According to Newton's second law, the acceleration of particle $i$, $\boldsymbol{a}_i \in \mathbb{R}^3$ is proportional to the aggregated force from other particles $\sum_{j \neq i} \boldsymbol{f}_{ij}$. All symbols will be specified with a superscript $t$ for temporal denotations, e.g., $\boldsymbol{x}_i^t$ indicating the position of particle $i$ at time $t$. In this paper, we are mainly concerned with the prediction task: we need to seek out a function $\phi(\{(\boldsymbol{S}_i^0, h_i^0)\}_{i=1}^N)$ given the initial states $\{\boldsymbol{S}_i^0\}_{i=1}^N$ and features $\{h_i^0\}_{i=1}^N$ to forecast the future states $\{\boldsymbol{S}_i^T\}_{i=1}^N$ at time $T$.

As presented in Introduction, two kinds of inductive biases have been explored previously. The first one is to apply the graph structure to capture the distribution of particle states and their interactions (Battaglia et al., 2016; Kipf et al., 2018), where, particularly, the particle states $\boldsymbol{S}_i^0$ (along with $h_i^0$) are as node features and the interaction forces $\boldsymbol{f}_{ij}^0$ as edge messages. In this way, the transition function $\phi$ boils down to a GNN model. The second inductive bias is that $\phi$ should be equivariant to any translation/reflection/rotation of the input states. By saying equivariance, we imply

$$\phi(\{(g \cdot \boldsymbol{S}_i^0, h_i^0)\}_{i=1}^N) = g \cdot \phi(\{(\boldsymbol{S}_i^0, h_i^0)\}_{i=1}^N), \forall g \in \mathcal{O}(3), \forall \boldsymbol{S}_i^0, \forall h_i^0. \tag{1}$$

Here, $\mathcal{O}(3)$ defines the 3D orthogonal group (Fuchs et al., 2020) that consists of translation, reflection and rotation transformations; $g \cdot \boldsymbol{S}_i^0$ denotes to perform transformation $g$ on the states $\boldsymbol{S}_i^0$, and it is instantiated as $\boldsymbol{R}(\boldsymbol{x}_i^0 + \boldsymbol{b})$ for the position and $\boldsymbol{R}\boldsymbol{v}_i^0$ for the velocity, where $\boldsymbol{R} \in \mathbb{R}^{3 \times 3}$ is the orthogonal matrix and $\boldsymbol{b} \in \mathbb{R}^3$ is the translation vector.

Several works (Thomas et al., 2018; Köhler et al., 2019; Fuchs et al., 2020; Satorras et al., 2021) have investigated both inductive biases, among which EGNN (Satorras et al., 2021) has achieved promising performance on N-body. The typical process in EGNN iterates the following steps:

---

[1]Although $h_i$ could be of multiple dimensions, we still call it as a non-vector value since the dimension of $h_i$ is distinct from that of the kinematics vector, for example $\boldsymbol{x}_i$, the latter of which has intrinsic geometry and should obey translation/rotation equivariance. This is also why we do not denote $h_i$ in bold.

$$a_i^l, h_i^l = \sum_j \varphi_{\text{egnn}}(x_{ji}^{l-1}, h_i^{l-1}, h_j^{l-1}, e_{ji}), \quad (2)$$

$$v_i^l = \psi(h_i^{l-1})v_i^{l-1} + a_i^l, \quad (3)$$

$$x_i^l = x_i^{l-1} + v_i^l, \quad (4)$$

where the superscript $l$ denotes the $l$-th layer; the acceleration $a_i^l$ returned by the message aggregation of $\varphi_{\text{egnn}}$ in Eq. 2 is adopted for the update of the velocity $v_i^l$ in Eq. 3 (multiplied by a scalar $\psi(h_i^{l-1}) \in \mathbb{R}$), followed by the renovation of the position $x_i^l$ in Eq. 4. The formulation of $\varphi_{\text{egnn}}$ is physically reasonable, since the interaction (actually the Coulomb force) between particles $i$ and $j$ truly depends on their relative position $x_{ji}^{l-1} = x_i^{l-1} - x_j^{l-1}$, node features $h_i^{l-1}$ and $h_j^{l-1}$, and edge feature $e_{ji}$. To enable equivariant message passing, EGNN has developed a specific form of $\varphi_{\text{egnn}}$ to let $a_i^l$ be equivariant and $h_i^l$ be invariant in terms of the input $x_{ji}^{l-1}$, which will be presented in Eq. 10. Notice that the computations for $f_i^l$ and $h_i^l$ are actually by two different functions, and we have abbreviated them into one in Eq. 2 (and also Eq. 5 later), since they share the same inputs; besides, their parameters are shared following EGNN.

## 3.1 OUR GENERAL ARCHITECTURE

Despite the desirable performance on N-body, existing methods are incapable of maintaining the geometric constraint. In this section, we design a general architecture that intrinsically meets the requirement of geometry constraints by making use of the generalized coordinates.

We define $\mathcal{O}_k = \{P_i\}_{i=1}^{n_k}$ a structural object composed of $n_k$ rigidly connected particles. Fig. 1 (b) illustrates two examples of the structural object, the stick with 2 connected particles and the hinge with 3 particles. To preserve the distance, the dynamics of the two particles on a stick should be updated in a joint way, rather than fulfilling the independent process in EGNN (Eq. 3 and 4). Besides, the force on each particle within $\mathcal{O}_k$ will indirectly influence the dynamics of others through the physical connections. This requires us to analyze the dynamics of the particles in $\mathcal{O}_k$ as a whole, which is implemented by generalized coordinates. There could be multiple generalized coordinates for each $\mathcal{O}_k$, some located in the Cartesian space but some in the angle space. For instance, the states of a stick can be decoupled by two independent sets of generalized coordinates: the state of particle 1 as the Cartesian coordinates and the relative rotation angles of particle 2 to 1 as the angle coordinates. For conciseness, this section only focus on the Cartesian part which essentially determines the local coordinates in $\mathcal{O}_k$, with providing full examples in § 3.3. We denote the position, velocity and acceleration of the generalized Cartesian coordinates as $q_k$, $\dot{q}_k$ and $\ddot{q}_k$, respectively.

We now detail how to update the states of $\mathcal{O}_k$. In Fig. 2, we first compute the interaction force between each particle and others, and aggregate information of all particles within each structural object to infer the acceleration of the generalized coordinates (which is termed as the generalized acceleration henceforth) by inverse dynamics. Then, the dynamical updates are carried out in the space of the generalized coordinates. Finally, the updated generalized coordinates will be projected back to the particles' states via the forward kinematics.

$$f_i^l, h_i^l = \sum_j \varphi_1(x_{ji}^{l-1}, h_i^{l-1}, h_j^{l-1}, e_{ji}), \quad (5)$$

$$\ddot{q}_k^l = \sum_{i \in \mathcal{O}_k} \varphi_2(f_i^l, x_{ki}^{l-1}, v_{ki}^{l-1}), \quad (6)$$

$$\dot{q}_k^l = \psi(\sum_{i \in \mathcal{O}_k} h_i^{l-1})\dot{q}_k^{l-1} + \ddot{q}_k^l, \quad (7)$$

$$q_k^l = q_k^{l-1} + \dot{q}_k^l, \quad (8)$$

$$x_i^l, v_i^l = \text{FK}(q_k^l, \dot{q}_k^l), \forall i \in \mathcal{O}_k, \quad (9)$$

Figure 2: The flowchart of our GMN.

where, the elements of the system can be described in two views, $\{\mathcal{O}_k\}_{k=1}^K$ as the object-level view and $\{P_i\}_{i=1}^N$ as the particle-level view; for distinction, we index the structural object with the subscript $k$ and particles with $i$ or $j$. We explain each equation separately.

**Interaction force (Eq. 5).** The interaction force $f_i^l$ is computed analogous to Eq. 2. EGNN

straightly regards the interaction force to be the acceleration of each particle in Eq. 2. But here, given the constraints between particles, we record the force as an intermediate variable that will contribute to the inference of the generalized acceleration in the next step.

**Inverse dynamics (Eq. 6).** This step is the core of our methodology. The generalized acceleration $\ddot{\boldsymbol{q}}_k^l$ is dependent to the forces $\boldsymbol{f}_i^l$ on all particles within $\mathcal{O}_k$, and their relative positions $\boldsymbol{x}_{ki}^{l-1} = \boldsymbol{x}_i^{l-1} - \boldsymbol{q}_k^{l-1}$ and relative velocities $\boldsymbol{v}_{ki}^{l-1} = \boldsymbol{v}_i^{l-1} - \dot{\boldsymbol{q}}_k^{l-1}$ with regard to the generalized coordinates. The formulation of Eq. 6 is physics-inspired. In Appendix (Eq. 26), we have analytically derived the dynamics of hinges, where the acceleration of the hinge is indeed related to the forces, relative positions and relative velocities of all particles in each hinge. This is reasonable in mechanics, since the cross product $\boldsymbol{x}_{ki}^{l-1} \times \boldsymbol{f}_i^l$ yields the torque, and the relative velocity $\boldsymbol{v}_{ki}^{l-1}$ is related to the centrifugal force of particle $i$ around $\boldsymbol{q}_k$, both of which influence the acceleration $\ddot{\boldsymbol{q}}_k$. Different from the analytical form which is always complex and hard to compute in practice, we will employ a learnable and equivariant function with universal expressivity. The details are in § 3.2.

**Generalized update (Eq. 7-8).** The updates of the generalized coordinates are akin to Eq. 3 and 4 in EGNN, as the dimensions of the generalized coordinates have been made independent. Notice that in Eq. 7 the scalar factor for $\dot{\boldsymbol{q}}_k^{l-1}$ takes as input the summation of all hidden features, which is a generalized form of Eq. 3 for multiple particles in $\mathcal{O}_k$.

**Forward kinematics (Eq. 9).** Once the generalized coordinates have been refreshed, we can derive the states of all particles in $\mathcal{O}_k$ by proceeding the forward kinematics. Different system of $\mathcal{O}_k$ could have different type of the forward kinematics. Here we denote it as the function $\mathrm{FK}(\cdot)$ in general, while providing the specifications in § 3.3.

Our method will reduce to EGNN if setting the generalized coordinates as the states of each particle and utilize the identify map in Eq. 6 and 9. Alg. 1 has summarized the updates for all objects.

## 3.2 Equivariant Message Passing

As shown in Eq. 1, the Euclidean equivariance on the estimation function $\phi$ is necessary for ensuring the physical symmetry. When considering this property in our case, we demand the interaction force (Eq. 5) and generalized acceleration (Eq. 6) to be equivariant with respect to orthogonal transformations, while other equations are already equivariant[2]. EGNN (Satorras et al., 2021) has developed a particular orthogonality-equivariant form for the acceleration output in Eq. 2:

$$\varphi_{\mathrm{egnn}}(\boldsymbol{x}, h) \coloneqq \boldsymbol{x}\sigma_w(\|\boldsymbol{x}\|_2^2, h), \tag{10}$$

where $\sigma_w(\cdot)$ is an arbitrary Multi-Layer Perceptron (MLP) with parameter $w$, and we have abbreviated other non-vector terms in Eq. 2 as $h$. This formulation does satisfy the rotation equivariance, but it is unknown if it can be generalized to functions (such as Eq. 6) with multiple input vectors, and more importantly, its representation completeness is never explored rigorously. In this section, we propose a general form of orthogonality-equivariant functions with necessary theoretical guarantees.

Without loss of generality, the target function we would like to enforce equivariance is denoted as $\varphi(\boldsymbol{Z}, h) : \mathbb{R}^{d \times m} \times \mathbb{R}^c \to \mathbb{R}^{d \times m'}$. We define the below formulation,

$$\varphi(\boldsymbol{Z}, h) \coloneqq \boldsymbol{Z}\sigma_w(\boldsymbol{Z}^\top \boldsymbol{Z}, h). \tag{11}$$

It is easy to justify the function $\varphi(\boldsymbol{Z}, h)$ in Eq. 11 is equivariant to any orthogonal matrix *i.e.*, $\varphi(\boldsymbol{O}\boldsymbol{Z}, h) = \boldsymbol{O}\varphi(\boldsymbol{Z}, h), \forall \boldsymbol{O} \in \mathbb{R}^{d \times d}, \boldsymbol{O}^\top \boldsymbol{O} = \boldsymbol{I}$. Apparently, Eq. 11 reduces to Eq. 10 by setting the number of vectors in $\boldsymbol{Z}$ as 1, namely, $m = m' = 1$. We immediately have the following theory.

**Theorem 1.** *If $m \geq d$ and the row rank of $\boldsymbol{Z}$ is full, i.e. rank$(\boldsymbol{Z}) = d$, then for any continuous orthogonality-equivariant function $\hat{\varphi}(\boldsymbol{Z}, h)$, there must exist an MLP $\sigma_w$ satisfying $\|\varphi(\boldsymbol{Z}, h) - \hat{\varphi}(\boldsymbol{Z}, h)\| < \epsilon$ for arbitrarily small error $\epsilon$.*

The proof employs the universality of MLP (Cybenko, 1989; Hornik, 1991), with the entire details deferred in Appendix. Theorem 1 is nutritive, as it characterizes the rich expressivity of formulating the equivariant function via Eq. 11. The condition holds in general for the function like Eq. 6 whose input $\boldsymbol{Z} = (\boldsymbol{f}_i, \boldsymbol{x}_{ki}, \boldsymbol{v}_{ki}) \in \mathbb{R}^{3 \times 3}$ is of full rank when the force, position and velocity expand the whole space. Indeed, this condition holds with probability 1, stated by the following corollary.

---

[2]The translation equivariance holds naturally as we use relative positions in Eq. 5, making the force, acceleration, velocity to be invariant with respect to translation. After the addition with the previous position in Eq. 8, the updated position $\boldsymbol{q}_k^l$ is translation equivariant.

**Corollary 1.** *Assume $m \geq d$, and also the entries of $\mathbf{Z}$ are drawn independently from a distribution that is absolutely continuous with respect to the Lebesgue measure in $\mathbb{R}$. Then, almost surely, the conclusion of Theorem 1 holds.*

When the condition $m \geq d$ is invalid, the universal approximation still maintains if restricted in the linear subspace expanded by the columns of $\mathbf{Z}$.

**Corollary 2.** *For any continuous orthogonality-equivariant function $\hat{\varphi}(\mathbf{Z}, h)$ located in the linear subspace expanded by the columns of $\mathbf{Z}$, the conclusion of Theorem 1 holds universally.*

Corollary 2 tells that the message passing $\varphi_{\text{egnn}}$ in Eq. 10 is not universally expressive since $m < d$, and can only fit the vectors parallel to $\boldsymbol{x}$ (*i.e.* the relative position). Yet, $\varphi_{\text{egnn}}$ is still physically complete for the Coulomb force that is oriented by the relative position between two particles.

In our experiments, we find that more stable performance is delivered by further adding the normalization term specifically when $m > 1$:
$$\varphi(\mathbf{Z}, h) := \mathbf{Z}\sigma_w(\mathbf{Z}^\top \mathbf{Z}/\|\mathbf{Z}^\top \mathbf{Z}\|_F, h), \tag{12}$$
where $\|\cdot\|_F$ computes the Frobenius norm. Notice that adding normalization does not change the conclusions in Theorem 1 and its two corollaries, since the norm is also a function of $\mathbf{Z}^\top \mathbf{Z}$ that can be approximated by MLP. We implement $\varphi_1$ in Eq. 5 and $\varphi_2$ in Eq. 6 by using the general formulation Eq. 12, where $\mathbf{Z} = \boldsymbol{x}_{ji}^{l-1}$ and $\mathbf{Z} = (\boldsymbol{f}_i^l, \boldsymbol{x}_{ki}^{l-1}, \boldsymbol{v}_{ki}^{l-1})$, respectively. Note that for the update of hidden feature $h_i^l$ in Eq. 5, we do not need equivariance but invariance, hence we set $h_i^l = \sigma_w(\mathbf{Z}^\top \mathbf{Z}/\|\mathbf{Z}^\top \mathbf{Z}\|_F, h_i^{l-1})$ by just keeping the invariant part in Eq. 12.

### 3.3 IMPLEMENTATIONS OF FORWARD KINEMATICS

**Implementation of hinges.** A hinge, as displayed in Fig.3 (a), consists of three particles 0, 1, 2, and two sticks 01 and 02. The freedom degrees of this system can be explained in this way: particle 0 moves freely, and particles 1 and 2 can only rotate round particle 0 owing to the length constraint by the two sticks. Hence, the generalized coordinates include the states of particle 0 denoted as $\boldsymbol{q}_0 \in \mathbb{R}^3$ and the rotation Euler angles of stick 01 as $\boldsymbol{\theta}_{01} \in \mathbb{R}^3$ and 02 as $\boldsymbol{\theta}_{02} \in \mathbb{R}^3$.

As $\boldsymbol{q}_0$ are Cartesian, the dynamical update follows directly Eq. 5-8. With the forces $\{\boldsymbol{f}_i^l\}_{i=0}^3$ estimated in Eq. 5 and acceleration $\ddot{\boldsymbol{q}}_0^l$ by Eq. 6, the angle acceleration of $\boldsymbol{\theta}_{01}$ is calculated as $\ddot{\boldsymbol{\theta}}_{01}^l = (\boldsymbol{x}_{01}^{l-1} \times (\boldsymbol{f}_1^l - \ddot{\boldsymbol{q}}_0^l))/\|\boldsymbol{x}_{01}^{l-1}\|^2$ according to rigid mechanics, where the numerator is the relative torque, the denominator is the moment of inertia under unit mass, and $\times$ means the cross product. Then the angle velocity is $\dot{\boldsymbol{\theta}}_{01}^l = \psi'(h_0^{l-1} + h_1^{l-1} + h_2^{l-1})\dot{\boldsymbol{\theta}}_{01}^{l-1} + \ddot{\boldsymbol{\theta}}_{01}^l$ similar to Eq. 7. We do not need to record $\boldsymbol{\theta}_{01}$, since

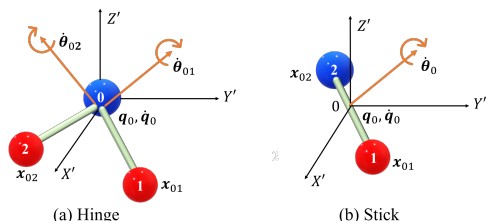

Figure 3: Illustrations of hinges and sticks.

the forward kinematics can be conducted without it. In detail, the forward kinematics in Eq. 9 is:
$$\boldsymbol{x}_1^l = \boldsymbol{q}_0^l + \text{rot}(\dot{\boldsymbol{\theta}}_{01}^l)\boldsymbol{x}_{01}^{l-1}, \quad \boldsymbol{v}_1^l = \dot{\boldsymbol{q}}_0^l + \dot{\boldsymbol{\theta}}_{01}^l \times \boldsymbol{x}_{01}^l, \tag{13}$$
where $\text{rot}(\dot{\boldsymbol{\theta}}_{01}^l)$ indicates the rotation matrix around the direction of $\dot{\boldsymbol{\theta}}_{01}^l$ by absolute angle $\|\dot{\boldsymbol{\theta}}_{01}^l\|$. The dynamic updates for particle 2 is similar. We put the whole details into Alg. 1 in Appendix.

**Implementation of sticks.** For a stick with particles 1 and 2 in Fig 3 (b), we choose the center as the generalized Cartesian coordinate $\boldsymbol{q}_0$, and the rotation of particle 1 $\boldsymbol{\theta}_{01}$ and particle 2 $\boldsymbol{\theta}_{02}$ ($\boldsymbol{\theta}_{01} = \boldsymbol{\theta}_{02}$) as the generalized angle coordinates. The dynamics propagation of $\boldsymbol{q}_0$ is given by Eq. 5-8. There are two choices to compute $\ddot{\boldsymbol{q}}_0^l$, one using the general form in Eq. 6 and the other one leveraging a simplified version as $\ddot{\boldsymbol{q}}_0^l = \sum_{i \in \mathcal{O}_k} \varphi(\boldsymbol{f}_i^l)$ by explicitly omitting the relative position and velocity. The physical motivation of introducing the simplified version is that the acceleration of a stick center is only affected by the forces according to the theorem of the motion of the center of mass. The angle accelerations are $\ddot{\boldsymbol{\theta}}_{01}^l = (\boldsymbol{x}_{01}^{l-1} \times \boldsymbol{f}_1^l + \boldsymbol{x}_{02}^{l-1} \times \boldsymbol{f}_2^l)/(\|\boldsymbol{x}_{01}^{l-1}\|^2 + \|\boldsymbol{x}_{02}^{l-1}\|^2)$, and the velocity becomes $\dot{\boldsymbol{\theta}}_{01}^l = \psi'(h_1^{l-1} + h_2^{l-1})\dot{\boldsymbol{\theta}}_{01}^{l-1} + \ddot{\boldsymbol{\theta}}_{01}^l$. The states of particles 1 and 2 are renewed as the same as Eq. 13.

**Learnable FK.** Besides the above hand-crafted FK, we propose a learnable variant by replacing Eq. (7-9) with the update $\boldsymbol{v}_i^l = \phi(h_i^{l-1})\boldsymbol{v}_i^{l-1} + \rho(\ddot{\boldsymbol{q}}_k^l, \boldsymbol{x}_{ki}^{l-1}, \boldsymbol{f}_i^l), \boldsymbol{x}_i^l = \boldsymbol{x}_i^{l-1} + \boldsymbol{v}_i^l$, where $\rho$ is the equivariant function via Eq. 12. Full details are provided in Appendix I.

Table 1: Prediction error ($\times 10^{-2}$) on various types of systems. The header of each column "$p, s, h$" denotes the scenario with $p$ isolated particles, $s$ sticks and $h$ hinges. Results averaged across 3 runs.

| | |Train| = 500 | | | | | |Train| = 1500 | | | | |
| --- | --- | --- | --- | --- | --- | --- | --- | --- | --- | --- |
| | 1,2,0 | 2,0,1 | 3,2,1 | 0,10,0 | 5,3,3 | 1,2,0 | 2,0,1 | 3,2,1 | 0,10,0 | 5,3,3 |
| Linear | 8.23±0.00 | 7.55±0.00 | 9.76±0.00 | 11.36±0.00 | 11.62±0.00 | 8.22±0.00 | 7.55±0.00 | 9.76±0.00 | 11.36±0.00 | 11.62±0.00 |
| GNN | 5.33±0.07 | 5.01±0.08 | 7.58±0.08 | 9.83±0.04 | 9.77±0.02 | 3.61±0.13 | 3.23±0.07 | 4.73±0.11 | 7.97±0.44 | 7.91±0.31 |
| TFN | 11.54±0.38 | 9.87±0.27 | 11.66±0.08 | 13.43±0.31 | 12.23±0.12 | 5.86±0.35 | 4.97±0.23 | 8.51±0.14 | 11.21±0.21 | 10.75±0.08 |
| SE(3)-Tr. | 5.54±0.06 | 5.14±0.03 | 8.95±0.04 | 11.42±0.01 | 11.59±0.01 | 5.02±0.03 | 4.68±0.05 | 8.39±0.02 | 10.82±0.03 | 10.85±0.02 |
| RF | 3.50±0.17 | 3.07±0.24 | 5.25±0.44 | 7.59±0.25 | 7.73±0.39 | 2.97±0.15 | 2.19±0.11 | 3.80±0.25 | 5.71±0.31 | 5.66±0.27 |
| EGNN | 2.81±0.12 | 2.27±0.04 | 4.67±0.07 | 4.75±0.05 | 4.59±0.07 | 2.59±0.10 | 1.86±0.02 | 2.54±0.01 | 2.79±0.04 | 3.25±0.07 |
| EGNNReg | 2.94±0.01 | 2.66±0.06 | 7.01±0.34 | 5.03±0.08 | 6.31±0.04 | 2.74±0.08 | 1.58±0.03 | 2.62±0.05 | 3.03±0.07 | 3.07±0.04 |
| GMN | 1.84±0.02 | 2.02±0.02 | 2.48±0.04 | 2.92±0.04 | 4.08±0.03 | 1.68±0.04 | 1.47±0.03 | 2.10±0.04 | 2.32±0.02 | 2.86±0.01 |

## 4 EXPERIMENTS

### 4.1 SIMULATION DATASET: CONSTRAINED N-BODY

**Datasets.** We inherit the 3D extension of Fuchs et al. (2020) based on the N-body simulation introduced in Kipf et al. (2018). For each trajectory, we provide the initial states of the system $\{S_i^0\}_{i=1}^N$, the particles' charges $\{c_i \in (-1, 1)\}_{i=1}^N$ and a configuration indicating which particles are connected by sticks or hinges. The task is to predict the final positions $\{x_i^T\}_{i=1}^N$ of the particles when $T = 1000$. The validation and testing sets contain 2000 trajectories. We evaluate the prediction error by the MSE metric. Compared with the simulation conducted in Fuchs et al. (2020); Satorras et al. (2021), our dataset is more challenging in three senses: **1.** We consider systems with multiple scales, including 5, 10, and 20 particles in total, respectively. **2.** We introduce to the system the dynamics of hinges and sticks (depicted in Appendix B), and construct various combinations between these objects. **3.** We investigate the performance of each model across different scales of training set, *e.g.*, 500 and 1500, to see how the models perform with scarce or relatively abundant training data. The system consisting of $p$ isolated particles, $s$ sticks and $h$ hinges, is abbreviated as $(p, s, h)$.

**Implementation details.** Following Satorras et al. (2021), we use a linear mapping of the scale of initial velocity $\|v_i^0\|_2$ as the input node feature $h_i^0$. The edge feature is provided by a concatenation of the product of charges $c_i c_j$ and an edge type indicator $I_{ij}$, where $I_{ij}$ is valued as 0 if node $i$ and $j$ are disconnected, 1 if connected by a stick, and 2 if connected by a hinge. Note that this edge type indicator is an augmentation over the original setting in EGNN, designed to enforce EGNN and other baselines the ability to distinguish different types of edges, namely, with or without constraints. Other settings including the hyper-parameters are introduced in Appendix E.

**Comparison with SOTAs.** Table 1 reports the performance of GMN and various compared models: EGNN (Satorras et al., 2021) and its regulated version EGNNReg, SE(3)-Transformer (Fuchs et al., 2020), Radial-Field (RF) (Köhler et al., 2019), Tensor-Field-Network (TFN) (Thomas et al., 2018), and other two baselines, GNN and the Linear prediction (Satorras et al., 2021). Regarding EGNN and EGNNReg, they share the same backbones (*i.e.* $\varphi_1$ and $\psi$) and training hyper-parameters (learning rates, layer number, etc) with our GMN for a fair comparison. For EGNNReg, we explicitly involve a regularization term during training by enforcing the geometrical constraints, namely preserving the lengths between two particles on sticks and hinges; the regularization factor is ranged from 0.01 to 0.1, where the value giving the best performance is selected. The default settings of SE(3)-Transformer and TFN perform poorly on our experiments, hence we have tried our best to tune their hyper-parameters by validation. From Table 1, we have these observations:

**1.** GMN achieves the best performance in all scenarios, suggesting its general superiority. **2.** GMN is more robust when the complexity of the system increases. On |Train| = 500, for example, the performance of GMN degenerates slightly by increasing the number of particles and hinges (*e.g.* from (1,2,0) to (3,2,1)), while other methods (such as EGNN) drops significantly. **3.** Reducing the training size will hinder the performance of all compared methods remarkably. On the contrary, GMN still performs promisingly in general. For instance, on (3,2,1), EGNNReg becomes much worse from 2.62 to 7.01 when the volume of training data is decreased from 1500 to 500, whereas the change of GMN is smaller (2.10 v.s. 2.48). This is reasonable as GMN has explicitly encoded the constraints as opposed to EGNN and EGNNReg that learn to remember constraints by training.

Table 2: Generalization across different systems. All models are trained in the (3,2,1) scenario.

| | |Train| = 500 | | | |Train| = 1500 | | |
| | 3,2,1 | 2,4,0 | 1,0,3 | Average | 3,2,1 | 2,4,0 | 1,0,3 | Average |
|---|---|---|---|---|---|---|---|---|
| GNN | 7.58 | 8.06 | 8.37 | 8.00 | 4.73 | 4.98 | 5.58 | 5.10 |
| EGNN | 4.67 | 3.42 | 4.40 | 4.16 | 2.54 | 2.75 | 3.49 | 2.93 |
| EGNNReg | 7.01 | 4.49 | 6.62 | 6.04 | 2.62 | 2.62 | 4.29 | 3.18 |
| GMN | **2.48** | **2.53** | **3.28** | **2.76** | **2.10** | **2.18** | **2.65** | **2.31** |

**Data efficiency.** Fig. 4 depicts the prediction errors on (3,2,1) when the training size varies. It is observed that GMN acts steadily, justifying its benefit in data efficiency. Once again, both EGNN and EGNNReg deliver much worse performance when the training size is small, and they approach GMN when the training dataset is enlarged sufficiently. In physics, it is important to ensure the physics rules that are discovered by a relatively small number of experiments to be general enough for explaining universal phenomena. Hence, data efficiency, as a key advantage of GMN, comes as an important require-

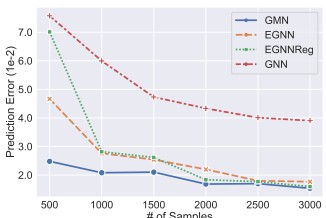

Figure 4: Data efficiency.

ment for learning to model physical systems. Besides, by the comparison between GNN and other equivariant models, it is seen that equivariance is a crucial point for performance guarantee.

**The generalization capability across different systems.** It is interesting to check how the models perform when trained on one system but tested on others. Hence, Table 2 tests the generalization from (3,2,1) to (2,4,0) and (1,0,3). Interestingly, the performances of all models on new systems are comparable with the original environment. We conjecture that this ability could be attributed to the usage of GNN in capturing the combination diversity of the objects. As before, GMN performs best.

**Ablation studies. 1.** The function $\varphi_2$ is designed to be equivariant in Eq. 12. To justify this necessity, we replace $\varphi_2$ with an MLP of the same size, and report the errors in Table 3, from which we confirm that removing equivariance incurs detriment to the performance. **2.** The default setting of GMN in Eq. 6 is leveraging unshared acceleration inference $\varphi_2(\boldsymbol{f}_i^l)$ for sticks and $\varphi_2(\boldsymbol{f}_i^l, \boldsymbol{x}_{ki}^{l-1}, \boldsymbol{v}_{ki}^{l-1})$ for hinges. Here

Table 3: Ablations. "O.F." denotes numerical over-flow.

| | |Train| = 500 | | |Train| = 1500 | |
| | 3,2,1 | 5,3,3 | 3,2,1 | 5,3,3 |
|---|---|---|---|---|
| GMN | **2.48** | **4.08** | **2.10** | **2.86** |
| GMN-L | 3.19 | 4.34 | 2.28 | 3.03 |
| w/o Equivariance | 3.74 | 4.41 | 2.46 | 3.29 |
| $\varphi_2$ with $(\boldsymbol{f}_i^l, \boldsymbol{x}_{ki}^{l-1}, \boldsymbol{v}_{ki}^{l-1})$ | 2.86 | 4.15 | 2.20 | 3.00 |
| $\varphi_2$ with $(\boldsymbol{f}_i^l, \boldsymbol{x}_{ki}^{l-1}, \boldsymbol{v}_{ki}^{l-1})$, shared | 3.86 | 4.25 | 2.30 | 3.10 |
| $\varphi_2$ with only $\boldsymbol{f}_i^l$ | 3.10 | 4.39 | 2.34 | 4.19 |
| $\varphi_2$ with only $\boldsymbol{f}_i^l$, shared | 2.91 | 4.94 | 2.39 | 3.45 |
| w/o Normalization | 3.15 | O.F. | O.F. | O.F. |

we investigate different cases by assigning the identical form of $\varphi_2$ for sticks and hinges with shared or unshared parameters. Applying identical $\varphi_2(\boldsymbol{f}_i^l, \boldsymbol{x}_{ki}^{l-1}, \boldsymbol{v}_{ki}^{l-1})$ mostly outperforms the case by using $\varphi_2(\boldsymbol{f}_i^l)$, probably owing to the better expressivity of the former version. Yet, both cases are worse than our design, implying that the dynamics of sticks and hinges should be modeled distinctly. **3.** We have introduced a normalization term in Eq. 12. Table 3 demonstrates that eliminating this term leads to divergence during training, possibly owing to the numerical instability in the forward/backward propagation. **4.** We have also implemented GMN-L that replaces the hand-crafted FK in GMN with a learnable black-box equivariant function. GMN-L outperforms EGNN in various settings, verifying the validity of using our proposed equivariant message passing layer and the object-level message $\ddot{\boldsymbol{q}}_k$ in Eq. 6. Yet, GMN-L still yields minor gap with GMN, implying the benefit of involving domain knowledge. The full results are deferred to Appendix I.

## 4.2 APPLICATIONS ON REAL-WORLD DATASETS

This subsection introduces how to apply GMN to practical applications including MD17 (Chmiela et al., 2017) and CMU Motion Capture (CMU, 2003). It is not required to manually derive the entire kinematics for these complex systems; instead, each input system is decomposed as a set of particles and sticks (*e.g.*, the circles in Fig. 5), which can be directly processed by our current formulation of GMN without any modification. The core is modeling partial length-constraints of the input system with disjoint sticks. The full details of the kinematics decomposition trick are in Appendix C.

Table 4: Prediction error ($\times 10^{-2}$) on MD17 dataset. Results averaged across 3 runs.

| | Aspirin | Benzene | Ethanol | Malonaldehyde | Naphthalene | Salicylic | Toluene | Uracil |
|---|---|---|---|---|---|---|---|---|
| RF | $10.94_{\pm 0.01}$ | $103.72_{\pm 1.29}$ | $\underline{4.64}_{\pm 0.01}$ | $13.93_{\pm 0.03}$ | $0.50_{\pm 0.01}$ | $1.23_{\pm 0.01}$ | $10.93_{\pm 0.04}$ | $\underline{0.64}_{\pm 0.01}$ |
| TFN | $12.37_{\pm 0.18}$ | $58.48_{\pm 1.98}$ | $4.81_{\pm 0.04}$ | $13.62_{\pm 0.08}$ | $0.49_{\pm 0.01}$ | $1.03_{\pm 0.02}$ | $10.89_{\pm 0.01}$ | $0.84_{\pm 0.02}$ |
| SE(3)-Tr. | $11.12_{\pm 0.06}$ | $68.11_{\pm 0.67}$ | $4.74_{\pm 0.13}$ | $13.89_{\pm 0.02}$ | $0.52_{\pm 0.01}$ | $1.13_{\pm 0.02}$ | $10.88_{\pm 0.06}$ | $0.79_{\pm 0.02}$ |
| EGNN | $14.41_{\pm 0.15}$ | $62.40_{\pm 0.53}$ | $\underline{4.64}_{\pm 0.01}$ | $13.64_{\pm 0.01}$ | $0.47_{\pm 0.02}$ | $1.02_{\pm 0.02}$ | $11.78_{\pm 0.07}$ | $\underline{0.64}_{\pm 0.01}$ |
| EGNNReg | $13.82_{\pm 0.19}$ | $61.68_{\pm 0.37}$ | $6.06_{\pm 0.01}$ | $13.49_{\pm 0.06}$ | $0.63_{\pm 0.01}$ | $1.68_{\pm 0.01}$ | $11.05_{\pm 0.01}$ | $0.66_{\pm 0.01}$ |
| GMN | $\underline{10.14}_{\pm 0.03}$ | $\mathbf{48.12}_{\pm 0.40}$ | $4.83_{\pm 0.01}$ | $\underline{13.11}_{\pm 0.03}$ | $\mathbf{0.40}_{\pm 0.01}$ | $\underline{0.91}_{\pm 0.01}$ | $\mathbf{10.22}_{\pm 0.08}$ | $\mathbf{0.59}_{\pm 0.01}$ |
| GMN-L | $\mathbf{9.76}_{\pm 0.11}$ | $\underline{54.17}_{\pm 0.69}$ | $\mathbf{4.63}_{\pm 0.01}$ | $\mathbf{12.82}_{\pm 0.03}$ | $\underline{0.41}_{\pm 0.01}$ | $\mathbf{0.88}_{\pm 0.01}$ | $\underline{10.45}_{\pm 0.04}$ | $\mathbf{0.59}_{\pm 0.01}$ |

**MD17.** We adopt MD17 (Chmiela et al., 2017) which involves the trajectories of eight molecules generated via molecular dynamics simulation. Our goal here is to predict the future positions of the atoms given the current system state. We observe that the lengths of chemical bonds remain very stable during the simulation, making it reasonable to model the bonds as sticks. The complete implementation details are deferred to Appendix E. As presented in Table 4, GMN outperforms other competitive equivariant models on 7 of the 8 molecules. Particu-

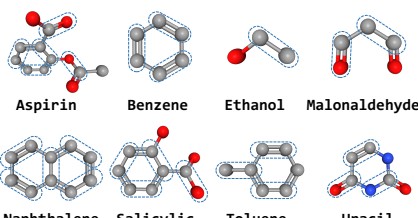

Figure 5: Molecules in MD17.

larly, on molecules with complex structures (*e.g.*, Aspirin, Benzene, and Salicylic), the improvement of GMN is more significant, showcasing the benefit of constraint modeling on the bonds. Yet, we also observe that the constraint-aware models (GMN and EGNNReg) perform worse than others on Ethanol, possibly because Ethanol is a relatively small molecule with simple structure, where considering the bond constraints possibly makes less benefit but instead hinders the learning. Surprisingly, GMN-L showcases very competitive performance on this dataset. It surpasses GMN on 4 of the 8 molecules, exhibiting that learnable FK works in practice even it does not involve domain knowledge of the constraint into kinematics modeling.

**CMU Motion Capture.** We use the motion data from the CMU Motion Capture Database (CMU, 2003), which contains the trajectory of human motion in various scenarios. Different parts of the human body could be treated as hard rigid-body constraints. We focus on the walking motion of single object (subject #35) containing 23 trials, similar to Kipf et al. (2018). As depicted in Table 5, GMN outperforms other models by a large margin, verifying the efficacy of our equivariant constraint module on modeling complex rigid bodies. We further provide a visualization in Fig. 6, where GMN predicts the motion accurately while EGNN yields larger error. Again, GMN-L, although is inferior to GMN, is better than other methods remarkably.

Table 5: Prediction error ($\times 10^{-2}$) on motion capture. Results averaged across 3 runs.

| GNN | TFN | SE(3)-Tr. | RF | EGNN | EGNNReg | GMN | GMN-L |
|---|---|---|---|---|---|---|---|
| $67.3_{\pm 1.1}$ | $66.9_{\pm 2.7}$ | $60.9_{\pm 0.9}$ | $197.0_{\pm 1.0}$ | $59.1_{\pm 2.1}$ | $59.5_{\pm 2.2}$ | $\mathbf{43.9}_{\pm 1.1}$ | $\underline{50.9}_{\pm 0.7}$ |

Figure 6: Left to Right: initial position, GMN, EGNN (all in blue). Ground truths are in red.

## 5 CONCLUSION

In this paper, we propose Graph Mechanics Networks (GMN) that are capable of characterising constrained systems of interacting objects. The core is making use of generalized coordinates, by which the constraints are implicitly and exactly encapsulated in the forward kinematics. To enable Euclidean symmetry, we have developed a general form of equivariant functions to simulate the interaction forces and backward dynamics, whose expressivity is theoretically justified. For the simulated systems with particles, sticks and hinges, GMN outperforms existing methods regarding prediction error, constraint satisfaction and data efficiency. Moreover, the evaluations on two real-world datasets support the generalization ability of GMN towards complex systems.

## 6 Reproducibility Statement

The complete proof of the theorems is provided in Appendix A. The hyper-parameters and other experimental details are provided in Appendix E.

Our code is available at: `https://github.com/hanjq17/GMN`.

## 7 Ethics Statement

The research in this paper does NOT involve any human subject, and our dataset is not related to any issue of privacy and can be used publicly. All authors of this paper follow the ICLR Code of Ethics (https://iclr.cc/public/CodeOfEthics).

### Acknowledgments

This work was jointly supported by the following projects: the Scientific Innovation 2030 Major Project for New Generation of AI under Grant NO. 2020AAA0107300, Ministry of Science and Technology of the People's Republic of China; the National Natural Science Foundation of China (No.62006137); Tencent AI Lab Rhino-Bird Visiting Scholars Program (VS2022TEG001); Beijing Academy of Artificial Intelligence (BAAI).

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

## A  THE PROOF OF THEOREM 1

In the following, we define the orthogonal group as $\mathcal{O}(d) = \{O \in \mathbb{R}^{d \times d} \mid O^\top O = OO^\top = I_d\}$
Prior to providing the proof for Theorem 1, we first list two necessary lemmas below.

**Lemma 1.** *For any two matrices $Z_1, Z_2 \in \mathbb{R}^{d \times m}$, we have this equivalence: $\exists O \in \mathcal{O}(d), OZ_1 = Z_2 \Leftrightarrow Z_1^\top Z_1 = Z_2^\top Z_2$.*

*Proof.* We only need to prove the "$\Leftarrow$" direction, as "$\Rightarrow$" holds clearly. Suppose the SVD decomposition of $Z_1$ as $Z_1 = U_1 S_1 V_1^\top$ with the left-singular matrix $U_1 \in \mathcal{O}(d)$, the singular diagonal matrix $S_1 \in \mathbb{R}^{d \times m}$, and the right-singular matrix $V_1 \in \mathcal{O}(m)$. Since $Z_1^\top Z_1 = Z_2^\top Z_2$, then there must exists a certain SVD decomposition of $Z_2$ that shares the same singular matrix and right-singular matrix with $Z_1$, implying that $Z_2 = U_2 S_1 V_1^\top$, where $U_2 \in \mathcal{O}(d)$. Hence, $Z_2 = U_2 S_1 V_1^\top = U_2 U_1^\top U_1 S_1 V_1^\top = U_2 U_1^\top Z_1$, which concludes the proof owing to the orthogonality of $U_2 U_1^\top$. $\qquad\square$

**Lemma 2.** *The function $f : \mathbb{R}^{d \times m} \to \mathbb{R}^{m'}$ is invariant on $\mathcal{O}(d)$, namely, $f(OZ) = f(Z), \forall O \in \mathcal{O}(d), \forall Z \in \mathbb{R}^{d \times m}$, if and only if there exists function $g : \mathbb{R}^{m \times m} \to \mathbb{R}^{m'}$ satisfying $f(Z) = g(Z^\top Z)$.*

*Proof.* The sufficiency is obvious. We now prove the necessity. We define the equivalence class $[Z_0] = \{Z \mid \exists O \in \mathcal{O}(d), OZ = Z_0\}$. Since $f$ is invariant to the orthogonal transformation, it means $f$ is actually a function on the equivalence class, *i.e.*, $f(Z) = f([Z])$. On the other hand, according to Lemma 1, we have $[Z_1] = [Z_2] \Leftrightarrow Z_1^\top Z_1 = Z_2^\top Z_2$, which implies the one-to-one correspondence between $[Z]$ and $Z^\top Z$; hence, there must exist a function $f'$ leading to $[Z] = f'(Z^\top Z)$, and $f'$ is continuous in terms of any invariant metric such as the norm $\|Z^\top Z\|$. Overall, $f(Z) = f([Z]) = f(f'(Z^\top Z)) := g(Z^\top Z)$. $\qquad\square$

We are now ready to prove Theorem 1 that is copied below for better readability.

**Theorem 1.** *If $m \geq d$ and the row rank of $Z$ is full,* i.e. *$rank(Z) = d$, then for any continuous orthogonality-equivariant function $\hat{\varphi}(Z, h)$, there must exist an MLP $\sigma_w$ satisfying $\|\varphi(Z, h) - \hat{\varphi}(Z, h)\| < \epsilon$ for arbitrarily small error $\epsilon$.*

*Proof.* Without loss of generality, we will omit the non-vector term $h$, which does not change the story but let our proof more concise. Because $Z$ is of full row-rank, the columns of an arbitrary function $\hat{\varphi}(Z)$ can be represented as a linear combination of the columns of $Z$; in other words, there must exist a function $\pi : \mathbb{R}^{d \times m} \to \mathbb{R}^{m \times m'}$ giving rise to $\hat{\varphi}(Z) = Z\pi(Z)$. Considering the orthogonality-equivariance, we derive the property of $\pi(Z)$ as:

$$\hat{\varphi}(OZ) = O\hat{\varphi}(Z),$$
$$\Rightarrow OZ\pi(OZ) = OZ\pi(Z),$$
$$\Rightarrow Z\pi(OZ) = Z\pi(Z). \tag{14}$$

Since $d \leq m$ and the row-rank of $Z$ is full, we perform the compact SVD decomposition on $Z$, namely, $Z = U_Z S_Z V_Z^\top$, where $U_Z \in \mathcal{O}(d)$, $\mathbb{R}^{d \times d} \ni S_Z > 0$, and $V_Z \in \mathbb{R}^{m \times d}$ satisfying $V_Z^\top V_Z = I_d$. Eq. 14 becomes:

$$Z\pi(OZ) = Z\pi(Z),$$
$$\Rightarrow U_Z S_Z V_Z^\top \pi(OZ) = U_Z S_Z V_Z^\top \pi(Z),$$
$$\Rightarrow S_Z V_Z^\top \pi(OZ) = S_Z V_Z^\top \pi(Z). \tag{15}$$

Given that $S_Z = \text{Eigen}(Z^\top Z)$ and $V_Z = \text{EigenVector}(Z^\top Z)$ are respectively the eigenvalues and eigenvectors of $Z^\top Z$ and are clearly invariant to the orthogonal transformation of $Z$. Their values can be numerically approximated by iterative programs, such as the power method (Mises & Pollaczek-Geiringer, 1929), thus can be treated as the continuous functions of $Z^\top Z$. Let us define $g'(Z) := S_Z V_Z^\top \pi(Z)$. Then, we keep deriving Eq. 15 by:

$$S_Z V_Z^\top \pi(OZ) = S_Z V_Z^\top \pi(Z),$$
$$\Rightarrow S_{OZ} V_{OZ}^\top \pi(OZ) = S_Z V_Z^\top \pi(Z),$$
$$\Rightarrow g'(OZ) = g'(Z). \tag{16}$$

According to Lemma 2, the function $g$ satisfies Eq. 16 if and only if it is written as $g'(\boldsymbol{Z}) = g(\boldsymbol{Z}^\top \boldsymbol{Z})$ for a certain function $g$. By checking the formulation of $\hat{\varphi}(\boldsymbol{Z})$ as demonstrated before, we arrive at

$$
\begin{aligned}
\hat{\varphi}(\boldsymbol{Z}) &= \boldsymbol{Z}\pi(\boldsymbol{Z}), \\
&= \boldsymbol{U_Z S_Z V_Z^\top} \pi(\boldsymbol{Z}), \\
&= \boldsymbol{U_Z} g'(\boldsymbol{Z}), \\
&= \boldsymbol{U_Z} g(\boldsymbol{Z}^\top \boldsymbol{Z}), \\
&= \boldsymbol{U_Z S_Z V_Z^\top V_Z S_Z^{-1}} g(\boldsymbol{Z}^\top \boldsymbol{Z}), \\
&= \boldsymbol{Z V_Z S_Z^{-1}} g(\boldsymbol{Z}^\top \boldsymbol{Z}), \\
&:= \boldsymbol{Z} \eta(\boldsymbol{Z}^\top \boldsymbol{Z}).
\end{aligned}
\tag{17}
$$

Here the function $\eta$ can be approximated by MLP whose universality has been justified by (Cybenko, 1989; Hornik, 1991). The conclusion of Theorem 1 is proved.

$\square$

**Corollary 1.** *Assume $m \geq d$, and also the entries of $\boldsymbol{Z}$ are drawn independently from a distribution that is absolutely continuous with respect to the Lebesgue measure in $\mathbb{R}$. Then, almost surely, the conclusion of Theorem 1 holds.*

*Proof.* This is straightforward. When $\text{rank}(\boldsymbol{Z}) < d$, the columns of $\boldsymbol{Z}$ are located in a subspace of $\mathbb{R}^d$ (for example a line or a plane in the 3D space), whose measure is zero. Therefore, the probability for making $\text{rank}(\boldsymbol{Z}) = d$ is 1, and we almost surely have the same conclusion as Theorem 1. $\square$

**Corollary 2.** *For any continuous orthogonality-equivariant function $\hat{\varphi}(\boldsymbol{Z}, h)$ whose output is located in the linear subspace expanded by the columns of $\boldsymbol{Z}$, the conclusion of Theorem 1 holds universally.*

*Proof.* According to the definition of $\hat{\varphi}(\boldsymbol{Z})$, we still obtain $\hat{\varphi}(\boldsymbol{Z}) = \boldsymbol{Z}\pi(\boldsymbol{Z})$. Let us assume $d > m$ (otherwise we can directly obtain Theorem 1), then the full SVD decomposition of $\boldsymbol{Z}$ is $\boldsymbol{Z} = \boldsymbol{U_Z S_Z V_Z^\top}$, with $\boldsymbol{U_Z} \in \mathcal{O}(d)$, $\boldsymbol{S_Z} \in \mathbb{R}^{d \times m}$, and $\boldsymbol{V_Z} \in \mathcal{O}(m)$. But here, $\boldsymbol{S_Z}$ is not strictly positive. Suppose $\boldsymbol{S_Z} = \begin{pmatrix} \boldsymbol{S_+} \\ \boldsymbol{0} \end{pmatrix}$, where $\boldsymbol{S_+} > 0$. We retain that $\boldsymbol{S_Z V_Z^\top} \pi(\boldsymbol{Z}) = g(\boldsymbol{Z}^\top \boldsymbol{Z})$ by imitating the proof in Eq. 14-16. Analogous to Eq. 17, we derive,

$$
\begin{aligned}
\hat{\varphi}(\boldsymbol{Z}) &= \boldsymbol{Z}\pi(\boldsymbol{Z}), \\
&= \boldsymbol{U_Z} g(\boldsymbol{Z}^\top \boldsymbol{Z}), \\
&= \boldsymbol{U_Z S_Z V_Z^\top V_Z} \left( \boldsymbol{S_+^{-1}}, \boldsymbol{0} \right) g(\boldsymbol{Z}^\top \boldsymbol{Z}), \\
&= \boldsymbol{Z V_Z} \left( \boldsymbol{S_+^{-1}}, \boldsymbol{0} \right) g(\boldsymbol{Z}^\top \boldsymbol{Z}), \\
&:= \boldsymbol{Z} \eta(\boldsymbol{Z}^\top \boldsymbol{Z}),
\end{aligned}
\tag{18}
$$

which concludes the proof.

$\square$

## B  THE DYNAMICS ANALYSES FOR STICKS AND HINGES

The analytical forms of the dynamics for sticks and hinges can be found from a mechanics book. Here we derive the formulas on our own to make our paper more self-contained. For simplicity, we consider all particles to be of the equal mass and the sticks of no mass.

**Dynamics analysis of sticks.** In Fig. 7 (Left), we assume the forces acting on particles 1 and 2 are separately $\boldsymbol{f}_1$ and $\boldsymbol{f}_2$. By following the theorem of the motion of the center of mass, the acceleration of the center is given by

$$
\ddot{\boldsymbol{q}}_0 = \frac{\boldsymbol{f}_1 + \boldsymbol{f}_2}{m}.
\tag{19}
$$

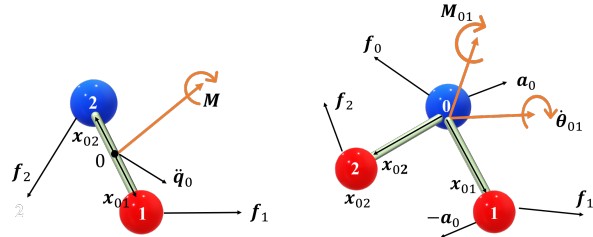

Figure 7: Left: Dynamics of sticks. Right: Dynamics of hinges.

The rotation accelerations of particles 1 and 2 around the center are calculated by

$$\ddot{\boldsymbol{\theta}}_{01} = \ddot{\boldsymbol{\theta}}_{02} = \frac{\boldsymbol{M}}{J} = \frac{\boldsymbol{x}_{01} \times \boldsymbol{f}_1 + \boldsymbol{x}_{02} \times \boldsymbol{f}_2}{m\|\boldsymbol{x}_{01}\|^2 + m\|\boldsymbol{x}_{02}\|^2}, \tag{20}$$

where $\boldsymbol{M}$ defines the total torque and $J$ is the moments of inertia.

**Dynamics analysis of hinges.** The analysis for hinges is more complicated than sticks. In Fig. 7 (Right), the forces on particles 0, 1 and 2 are $\boldsymbol{f}_0$, $\boldsymbol{f}_1$ and $\boldsymbol{f}_2$. By using Newton's second law,

$$\boldsymbol{a}_0 + \boldsymbol{a}_1 + \boldsymbol{a}_2 = \frac{\boldsymbol{f}}{m}, \tag{21}$$

where the aggregated force is $\boldsymbol{f} = \boldsymbol{f}_0 + \boldsymbol{f}_1 + \boldsymbol{f}_2$. In addition, the kinematics relations between the three particles show that

$$\boldsymbol{a}_1 = \boldsymbol{a}_0 + \ddot{\boldsymbol{\theta}}_{01} \times \boldsymbol{x}_{01} + \dot{\boldsymbol{\theta}}_{01} \times \boldsymbol{\nu}_{01}, \tag{22}$$

$$\boldsymbol{a}_2 = \boldsymbol{a}_0 + \ddot{\boldsymbol{\theta}}_{02} \times \boldsymbol{x}_{02} + \dot{\boldsymbol{\theta}}_{02} \times \boldsymbol{\nu}_{02}, \tag{23}$$

where $\dot{\boldsymbol{\theta}}_{01}$ and $\ddot{\boldsymbol{\theta}}_{01}$ denote the speed and acceleration of the rotation angle of particle 1 around 0, and $\boldsymbol{\nu}_{01}$ is the corresponding linear velocity; the symbols $\dot{\boldsymbol{\theta}}_{02}$, $\ddot{\boldsymbol{\theta}}_{02}$ and $\boldsymbol{\nu}_{02}$ are defined similarly. Moreover, the relative rotation acceleration of particle 1 to 0 is caused by the external force $\boldsymbol{f}_1$ and the inertia force $-m\boldsymbol{a}_0$ acting on 1, which derives that

$$\ddot{\boldsymbol{\theta}}_{01} = \frac{\boldsymbol{M}_{01}}{J_{01}} = \frac{\boldsymbol{x}_{01} \times (\boldsymbol{f}_1 - m\boldsymbol{a}_0)}{m\|\boldsymbol{x}_{01}\|^2}. \tag{24}$$

Analogously,

$$\ddot{\boldsymbol{\theta}}_{02} = \frac{\boldsymbol{M}_{02}}{J_{02}} = \frac{\boldsymbol{x}_{02} \times (\boldsymbol{f}_2 - m\boldsymbol{a}_0)}{m\|\boldsymbol{x}_{02}\|^2}. \tag{25}$$

After substituting Eq. 24 into Eq. 22, and Eq. 25 into Eq. 23, and then rearranging Eq. 21, we have

$$\boldsymbol{a}_0 = (\boldsymbol{I} + \boldsymbol{e}_{01}\boldsymbol{e}_{01}^\top + \boldsymbol{e}_{02}\boldsymbol{e}_{02}^\top)^{-1}\boldsymbol{a}, \tag{26}$$

where $\boldsymbol{a} = \frac{\boldsymbol{f}}{m} - \dot{\boldsymbol{\theta}}_{01} \times \boldsymbol{\nu}_{01} - \dot{\boldsymbol{\theta}}_{02} \times \boldsymbol{\nu}_{02} - (\boldsymbol{I} - \boldsymbol{e}_{01}\boldsymbol{e}_{01}^\top)\frac{\boldsymbol{f}_1}{m} - (\boldsymbol{I} - \boldsymbol{e}_{02}\boldsymbol{e}_{02}^\top)\frac{\boldsymbol{f}_2}{m}$ with $\boldsymbol{e}_{01}$ and $\boldsymbol{e}_{02}$ represent the unit vectors along $\boldsymbol{x}_{01}$ and $\boldsymbol{x}_{02}$, respectively. We have applied a trick to simplify the derivation by observing $\frac{\boldsymbol{x}_{01} \times \boldsymbol{f}_1 \times \boldsymbol{x}_{01}}{\|\boldsymbol{x}_{01}\|^2} = (\boldsymbol{I} - \boldsymbol{e}_{01}\boldsymbol{e}_{01}^\top)\boldsymbol{f}_1$. Besides, $\boldsymbol{I} + \boldsymbol{e}_{01}\boldsymbol{e}_{01}^\top + \boldsymbol{e}_{02}\boldsymbol{e}_{02}^\top$ is invertible, hence Eq. 26 is always meaningful.

Substituting Eq. 26 back into Eq. 24 and Eq. 25 derives the values of $\ddot{\boldsymbol{\theta}}_{01}$ and $\ddot{\boldsymbol{\theta}}_{02}$. Note that in § 3.3, we employ the denotation of generalized coordinates by $\boldsymbol{q}_0$, $\dot{\boldsymbol{q}}_0$ and $\ddot{\boldsymbol{q}}_0$ which indeed share the same meaning with the Cartesian coordinates $\boldsymbol{x}_0$, $\boldsymbol{v}_0$ and $\boldsymbol{a}_0$ of particle 0.

## C  KINEMATICS DECOMPOSITION

We manually build the forward kinematics (Eq. 9) of the stick (and hinge), which relies on the domain knowledge of the underlying physics. However, for complex systems, we no longer require

Table 6: Prediction error ($\times 10^{-2}$) on various types of systems. The first column "$p, s, h$" denotes the scenario with p isolated particles, s sticks and h hinges. Models are trained with 500 samples.

|       | GMN  | EGNN | EGNNReg | GNN   | TFN   | SE(3)-Tr. | RF   | Linear |
|-------|------|------|---------|-------|-------|-----------|------|--------|
| 1,2,0 | 1.84 | 2.81 | 2.94    | 5.33  | 11.54 | 5.54      | 3.50 | 8.23   |
| 2,0,1 | 2.02 | 2.27 | 2.66    | 5.01  | 9.87  | 5.14      | 3.07 | 7.55   |
| 2,4,0 | 2.34 | 3.59 | 3.87    | 8.05  | 11.30 | 9.22      | 5.37 | 10.10  |
| 0,5,0 | 2.54 | 4.13 | 4.29    | 8.63  | 11.92 | 9.83      | 5.94 | 10.48  |
| 7,0,1 | 2.39 | 2.66 | 3.41    | 7.05  | 10.67 | 8.38      | 4.66 | 9.71   |
| 1,0,3 | 3.21 | 4.56 | 5.14    | 8.32  | 11.62 | 9.57      | 5.91 | 9.90   |
| 3,2,1 | 2.48 | 4.67 | 7.01    | 7.58  | 11.66 | 8.95      | 5.25 | 9.76   |
| 4,8,0 | 3.69 | 4.79 | 7.09    | 9.65  | 12.05 | 11.21     | 7.59 | 11.45  |
| 0,10,0| 2.92 | 4.75 | 5.03    | 9.83  | 13.43 | 11.42     | 7.59 | 11.36  |
| 8,0,4 | 3.37 | 4.17 | 5.32    | 9.49  | 11.72 | 11.12     | 7.51 | 11.44  |
| 2,0,6 | 4.06 | 5.06 | 5.58    | 10.13 | 12.13 | 11.74     | 8.15 | 11.61  |
| 5,3,3 | 4.08 | 4.59 | 6.31    | 9.77  | 12.23 | 11.59     | 7.73 | 11.62  |

to derive the kinematics of the entire system. Instead, we propose kinematics decomposition, a simple yet effective trick that can decompose each input system (an arbitrary graph) into particles and sticks. Taking the MD17 dataset as an example, for each molecule, we select certain bonds as sticks (the circles in Fig. 5) and the remaining atoms as isolated particles; in this way, we obtain a set of particles and sticks. Note that different sticks are not allowed to intersect; otherwise, it will generate two values for the intersecting particle of two sticks and cause ambiguity if these two values are distinct. Although this kind of kinematics decomposition will only maintain partial constraints, it greatly enlarges the application scope of our current formulation Eq. (5-9) without any revision. More importantly, our experiments verify that GMN by this formulation is sufficient to surpass other methods on complex systems like molecules. On CMU Motion Capture, since the motion graph contains no circle and is of the tree-like structure, it is tractable to derive the exact forward kinematics by recursive kinematics computation from the root node. Yet, we still encourage the usage of the above kinematics decomposition for its easy implementation and compatibility with our GMN.

## D   FULL ALGORITHMIC DETAILS

In the main body of the paper, for better readability, we first introduce the general pipeline of our method in § 3.1 and then present the implementation details by taking the angels into account in § 3.3. Here, we combine them into one singe algorithmic flowchart in Alg. 1.

## E   MORE EXPERIMENTAL DETAILS AND RESULTS

**Hyper-parameters and baselines.** For GNN, RF, EGNN, EGNNReg, and GMN, we empirically find that the following hyper-parameters generally work well, and use them across all experimental evaluations: batch size 200, Adam optimizer with learning rate 0.0005, hidden dim 64, and weight decay $1 \times 10^{-10}$. All models are evaluated with four layers. SE(3)-Transformer and TFN do not perform well on our datasets, potentially due to the challenge of highly complex and constrained systems. Consequently, we tune the hyper-parameters and adopt the following configuration: batch size 100, learning rate 0.001, hidden dim 64, representation degrees 3 and weight decay $1 \times 10^{-8}$. Models are trained for 600 epochs on the simulation dataset, and 500 epochs on the real-world datasets. EGNNReg is a variant of EGNN that explicitly adds the constraint error into its training loss by a regularization factor of $\lambda$. In our experiments, we also treat $\lambda$ as a hyper-parameter, and choose $\lambda$ that yields the best performance within the range [0.01, 0.1].

**Detailed experimental setup on MD17.** We randomly split the dataset into train/validation/test sets containing 500/2000/2000 frame pairs respectively. We choose $T = 5000$ as the span between the input and prediction frames, and the difference in positions as the input velocity. The hyper-

---

**Algorithm 1** Graph Mechanics Networks (GMNs)

---

**Input:** Initial states of all particles $\{S_i^0 = (x_i^0, v_i^0)\}_{i=1}^N$ and features $\{h_i^0\}_i^N$; Learnable equivariant functions $\varphi_1, \varphi_2, \psi, \psi'$; Layer number $L$.

Compute the generalized coordinates of all structural objects $\{(q_k^0, \dot{q}_k^0, \{\dot{\theta}_{ki}^0\}_{i \in \mathcal{O}_k})\}_{k=1}^K$.

**for** layer $l = 1$ to $L$ **do**

  **for** particle $i = 1$ to $N$ **do**

    Calculate the interaction force $f_i^l$ and feature $h_i^l$ for each particle by:

$$f_i^l, h_i^l = \sum_{j=1}^N \varphi_1(x_{ji}^{l-1}, h_i^{l-1}, h_j^{l-1}, e_{ji}). \tag{27}$$

  **end for**

  **for** object $k = 1$ to $K$ **do**

    Inference the generalized Cartesian acceleration $\ddot{q}_k^l$ via:

$$\ddot{q}_k^l = \sum_{i \in \mathcal{O}_k} \varphi_2(f_i^l), \quad \text{(for sticks)} \tag{28}$$

$$\ddot{q}_k^l = \sum_{i \in \mathcal{O}_k} \varphi_2(f_i^l, x_{ki}^{l-1}, v_{ki}^{l-1}). \quad \text{(for hinges)} \tag{29}$$

    Derive the generalized angle acceleration:

$$\ddot{\theta}_{ki} = \frac{\sum_{i \in \mathcal{O}_k} x_{ki} \times f_i}{\sum_{i \in \mathcal{O}_k} \|x_{ki}\|^2}, \forall i \in \mathcal{O}_k, \quad \text{(for sticks)} \tag{30}$$

$$\ddot{\theta}_{ki} = \frac{x_{ki} \times (f_i - \ddot{q}_k)}{\|x_{ki}\|^2}, \forall i \in \mathcal{O}_k. \quad \text{(for hinges)} \tag{31}$$

    Update the positions and velocities as follows:

$$\dot{q}_k^l = \psi(\sum_{i \in \mathcal{O}_k} h_i^{l-1})\dot{q}_k^{l-1} + \ddot{q}_k^l, \tag{32}$$

$$q_k^l = q_k^{l-1} + \dot{q}_k^l, \tag{33}$$

$$\dot{\theta}_{ki}^l = \psi'(\sum_{i \in \mathcal{O}_k} h_i^{l-1})\dot{\theta}_{ki}^{l-1} + \ddot{\theta}_{ki}^l, \forall i \in \mathcal{O}_k. \tag{34}$$

    Perform the forward kinematics for each particle in $\mathcal{O}_k$:

$$x_i^l = q_k^l + \text{rot}(\dot{\theta}_{ki}^l)x_{ki}^{l-1}, \forall i \in \mathcal{O}_k, \tag{35}$$

$$v_i^l = \dot{q}_k^l + \dot{\theta}_{ki}^l \times x_{ki}^l, \forall i \in \mathcal{O}_k. \tag{36}$$

  **end for**

**end for**

**Output:** The predicted states of all particles $\{S_i^L\}_{i=1}^N$.

---

parameters of all models are kept the same as the synthetic dataset. We masked out the hydrogen atoms, focusing on the prediction of large atoms. We further augment the original molecular graph with 2-hop neighbors, and concatenate the hop index with atom numbers of the connected atoms as well as the edge type indicator as the edge feature, similar to Shi et al. (2021). We use the norm of velocity concatenated with the atom number as the node feature. We randomly select bonds without commonly-connected atoms as sticks, and the rest of atoms as isolated particles. Although by this means not all the bond lengths are preserved, our experiment does illustrate that this is a simple but effective strategy of applying GMN on complex systems like molecules.

**Detailed experimental setup on CMU Motion Capture.** We first split the data into train/val/test sets containing 11/6/6 trials respectively. We then sample from the trials to get 200/600/600 frame

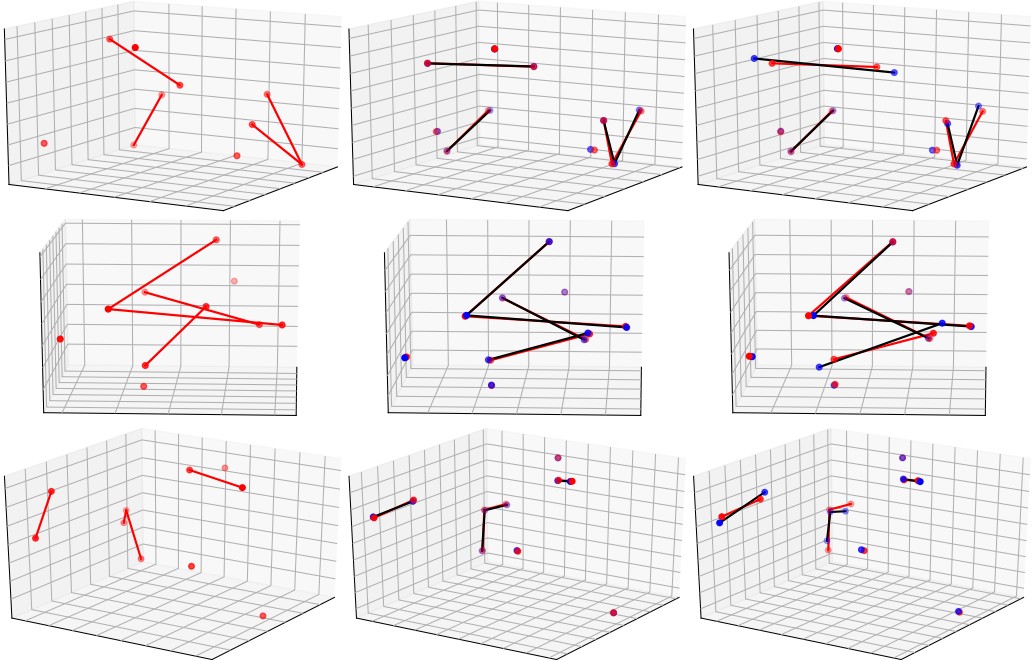

Figure 8: Left: initial position(s). Middle: the prediction(s) of GMN (in blue). Right: the prediction(s) of EGNN (in blue). Ground truth is marked in red. Better viewed by zooming in.

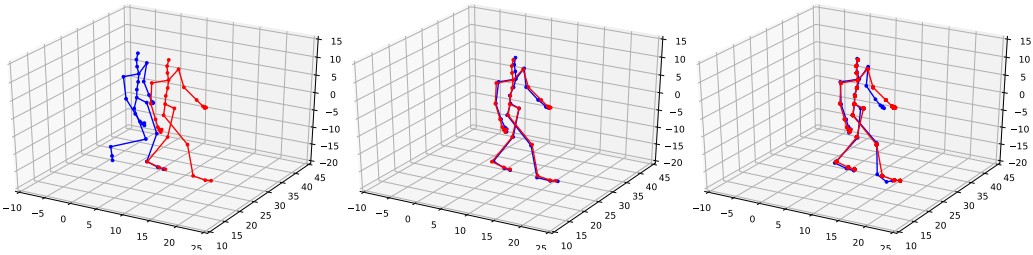

Figure 9: Left: initial position (in blue). Middle: the prediction of GMN (in blue). Right: the prediction of EGNN (in blue). Ground truths are marked in red. Better viewed by zooming in.

pairs with $T = 30$. We use the norm of velocity as node feature. We also augment the edges with 2-hop neighbors, and use the edge type indicator as edge feature. As for GMN, we sample 6 key bones of human body (as specified in Appendix C) as sticks, resulting in a system with 19 isolated particles and 6 sticks. Empirically, we select the edges connecting nodes (0, 11), (2, 3), (7, 8), (12, 13), (17, 18), and (24, 25) as sticks on the motion capture dataset, which indeed are the key parts of human body like the arms and legs.

**More visualizations.** We visualize the prediction outcomes by GMN and EGNN in Fig. 8 on the simulation dataset. GMN is found to be able to track the ground-truth trajectories accurately, whereas EGNN yields clear position errors and particularly breaks the constraints for sticks and hinges. These results are consistent with the performance in Table 1. Fig. 9 provides extra visualization on Motion Capture. Similarly, GMN yields more accurate prediction than EGNN.

**More experimental results.** In Table 6, we provide a comprehensive performance comparison of different models under more scenarios with 500 training samples. Clearly, GMN outperforms other models on all object combinations involved. Besides, we provide the learning curve for GMN, EGNN, and EGNNReg on the simulation dataset (3,2,1) in Fig. 10. GMN yields lower training loss and testing loss than EGNN and EGNNReg, benefiting from its constraint modeling.

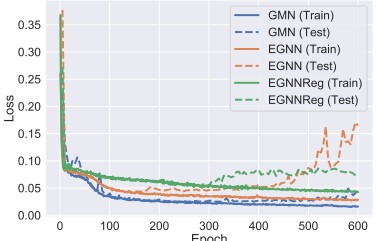

Figure 10: Learning curves on (3,2,1) with 500 training samples.

## F    MORE ABLATIONS

**Hinge treated as two sticks.** We treat 0-1, 0-2 as two sticks, and apply the stick FK respectively. Afterwards, we translate the two sticks such that the 0s coincide at the midpoint of their predicted positions. From Table 7, we find that this strategy performs worse than the hinge-modeled GMN, since splitting the hinge into two sticks overlooks the kinematics at the connected point. Yet and still, it performs better than EGNN, again verifying the benefit of our proposed constraint modeling.

Table 7: Ablation on hinge FK.

|  | |Train| = 500 | | | |Train| = 1500 | | |
| --- | --- | --- | --- | --- | --- | --- |
|  | 3,2,1 | 2,0,6 | 5,3,3 | 3,2,1 | 2,0,6 | 5,3,3 |
| EGNN | 4.67 | 5.06 | 4.59 | 2.54 | 3.50 | 3.42 |
| EGNNReg | 7.01 | 5.58 | 6.31 | 2.62 | 3.61 | 3.07 |
| GMN (Stick only) | 3.02 | 4.32 | 4.21 | 2.37 | 3.30 | 2.88 |
| GMN | 2.48 | 4.06 | 4.08 | 2.10 | 3.22 | 2.86 |

**Charges as node or edge feature.** In the experiment we by default assign $e_{ij} = c_i c_j$ as the edge feature (denoted as "Edge + C"). Here we instead concatenate $c_i$ to the node feature of particle $i$, and set all $e_{ij} = 0$ (denoted as "Node + C"). In Table 8 we observe that these alternatives on charges make very limited difference on performance, and indeed the models can learn the interaction of charges from node features, which is truly the case as depicted in Eq. 2 and 5.

Table 8: Comparison of charge-assigning strategies.

|  | Node + C | | | | | Edge + C | | | | |
| --- | --- | --- | --- | --- | --- | --- | --- | --- | --- | --- |
|  | 1,2,0 | 2,0,1 | 3,2,1 | 0,10,0 | 5,3,3 | 1,2,0 | 2,0,1 | 3,2,1 | 0,10,0 | 5,3,3 |
| EGNN | 2.89 | 2.28 | 4.25 | 4.80 | 4.50 | 2.81 | 2.27 | 4.67 | 4.75 | 4.59 |
| EGNNReg | 3.17 | 2.74 | 8.20 | 5.01 | 6.64 | 2.94 | 2.66 | 7.01 | 5.03 | 6.31 |
| GMN | 1.89 | 2.01 | 2.63 | 3.07 | 4.02 | 1.84 | 2.02 | 2.48 | 2.92 | 4.08 |

## G    CONSTRAINT SATISFACTION

**Constraint error.** The constraint error is computed as the total change in the lengths of sticks and hinges between the input and output, averaged per trajectory. Specifically, for hinges, the two edges are both considered.

**Results.** One vital feature of GMN is that it maintains the geometrical constraints exactly and inherently. To show this, Table 9 records the corresponding constraint errors of several typical models. The results do verify our claim that GMN always outputs near-zero errors (all below 1e-4). Although EGNNReg that augments EGNN with regulation helps in reducing the constraint errors, it is data-driven and limited by the number of training samples; further, the constraints are pursued softly, making it defective for the applications where hard constraints are indispensable.

Table 9: Constraint error on various types of systems.

|  | |Train| = 500 | | | | | |Train| = 1500 | | | |
| --- | --- | --- | --- | --- | --- | --- | --- | --- | --- | --- |
|  | 1,2,0 | 2,0,1 | 3,2,1 | 0,10,0 | 5,3,3 | 1,2,0 | 2,0,1 | 3,2,1 | 0,10,0 | 5,3,3 |
| GNN | 0.200 | 0.386 | 0.492 | 0.154 | 0.468 | 0.225 | 0.426 | 0.779 | 0.251 | 0.772 |
| EGNN | 0.220 | 0.370 | 0.714 | 0.248 | 0.760 | 0.217 | 0.317 | 0.521 | 0.139 | 0.596 |
| EGNNReg | 0.172 | 0.146 | 0.232 | 0.198 | 0.241 | 0.159 | 0.053 | 0.091 | 0.097 | 0.075 |
| GMN | **0.000** | **0.000** | **0.000** | **0.000** | **0.000** | **0.000** | **0.000** | **0.000** | **0.000** | **0.000** |

## H MORE DISCUSSIONS ON GENERALIZATION

In Table 2 we compare the generalization capability of GMN with other methods. Here, we denote GMN trained on (3,2,1) and tested across different scenarios as GMN-Transfer, and GMN trained and tested on the same dataset as GMN-Original. It is observed from Table 10 that GMN has strong generalization capability, since the transfer performance is very close to the original setting.

Table 10: Comparison of GMN in the transfer and original settings.

|  | |Train| = 500 | | | | |Train| = 1500 | | | |
| --- | --- | --- | --- | --- | --- | --- | --- | --- |
|  | 3,2,1 | 2,4,0 | 1,0,3 | Average | 3,2,1 | 2,4,0 | 1,0,3 | Average |
| GMN-Transfer | 2.48 | 2.53 | 3.28 | 2.76 | 2.10 | 2.18 | 2.65 | 2.31 |
| GMN-Original | 2.48 | 2.34 | 3.21 | 2.68 | 2.10 | 2.01 | 2.44 | 2.18 |

## I LEARNABLE FK

It is indeed instrumental to discuss whether a learnable black-box function, which requires less domain knowledge, could also yield competitive performance, and if our hand-crafted FK still shows advantage over the learnable counterpart. To answer these questions, we replace the hand-crafted part (Eq. (7-9) as well as the Euler angle computations in Sec. 3.1) with the following equations: $\boldsymbol{v}_i^l = \phi(h_i^{l-1})\boldsymbol{v}_i^{l-1} + \rho(\ddot{\boldsymbol{q}}_k^l, \boldsymbol{x}_{ki}^{l-1}, \boldsymbol{f}_i^l), \boldsymbol{x}_i^l = \boldsymbol{x}_i^{l-1} + \boldsymbol{v}_i^l$, where $\rho$ is the equivariant message passing layer we propose in Sec. 3.2. By this design, the parameterized FK preserves its equivariant property (and the theoretical universality), and compared to EGNN, it additionally leverages the information from the object-level generalized coordinates $\ddot{\boldsymbol{q}}_k^l$. We denote this variant of GMN as GMN-L. Moreover, since the parameterized FK inevitably loses the constraint-preserving property compared with the exact FK, therefore we also augment it with explicit constraint regularization, akin to what we did to EGNNReg. We hence denote this variant as GMN-LReg.

We evaluate the performance of GMN-L, GMN-LReg and compare them with GMN with exact FK as well as EGNN and EGNNReg in Table 11. We interestingly find that GMN-L consistently outperforms EGNN in various settings (as well as the regularized version), which again verifies both the validity of our proposed equivariant message passing layer and the efficacy of leveraging object-level message (i.e., $\ddot{\boldsymbol{q}}_k^l$) for the inference of FK. At the same time, GMN-L and GMN-LReg yield a minor gap with GMN, showing the evidence that hard-coding the constraints replaces a nontrivial amount of learning complexity.

Table 11: Comparison with GMN-L and GMN-LReg.

|  | |Train| = 500 | | | | | |Train| = 1500 | | | |
| --- | --- | --- | --- | --- | --- | --- | --- | --- | --- | --- |
|  | 1,2,0 | 2,0,1 | 3,2,1 | 0,10,0 | 5,3,3 | 1,2,0 | 2,0,1 | 3,2,1 | 0,10,0 | 5,3,3 |
| EGNN | 2.81 | 2.27 | 4.67 | 4.75 | 4.59 | 2.59 | 1.86 | 2.54 | 2.79 | 3.25 |
| EGNNReg | 2.94 | 2.66 | 7.01 | 5.03 | 6.31 | 2.74 | 1.58 | 2.62 | 3.03 | 3.07 |
| GMN-L | 2.32 | 2.09 | 3.19 | 3.88 | 4.34 | 1.93 | 1.56 | 2.28 | 2.72 | 3.03 |
| GMN-LReg | 2.52 | 2.23 | 3.34 | 3.67 | 4.31 | 1.91 | 1.88 | 2.49 | 2.61 | 3.00 |
| GMN | 1.84 | 2.02 | 2.48 | 2.92 | 4.08 | 1.68 | 1.47 | 2.10 | 2.32 | 2.86 |

## J    ROBUSTNESS TO ERRORS IN THE PHYSICAL PRIOR OF CONSTRAINTS

It is interesting to test the robustness of our model w.r.t. the noisy constraints. This scenario would sometimes arise in real-world scenarios where we might not be certain about the exact connectivity of the rigid body, and thus would involve slight errors in domain expertise. Since our paper focuses on the distance constraint other than the angle constraint, the following investigations will only involve noise into the stick connectivity. We design three random perturbation operations on the input rigid body prior: **1.** (Join) randomly selecting 2 isolated particles and joining them as if there is a stick connecting; **2.** (Split) randomly selecting an existing stick and splitting it as two isolated particles; **3.** (Join + Split) conducting operation 1 and 2 at the same time; and **4.** (Change in Length) randomly adding Gaussian noise $\mathcal{N}(0, 0.1L)$ to the length of a stick, where $L$ is its original length. Note that the operation is conducted independently for every training sample each time it is fed into the network.

We summarize the results in Table 12. We observe that these perturbations, although somehow hinder the performance, in general do not jeopardize the performance too much (difference in MSE $\leq 0.30$), indicating that GMN is not sensitive to slight errors of the input physical prior of the constraints and it is still able to learn to some degree of given the wrong constraints.

Table 12: Robustness test in various scenarios.

|  | |Train| = 500 | | | |Train| = 1500 | | |
| --- | --- | --- | --- | --- | --- | --- |
|  | 3,2,1 | 5,3,3 | 8,6,0 | 3,2,1 | 5,3,3 | 8,6,0 |
| GMN | 2.48 | 4.08 | 2.84 | 2.10 | 2.86 | 2.22 |
| GMN w/ Join | 2.59 | 4.27 | 2.98 | 2.22 | 3.16 | 2.37 |
| GMN w/ Split | 2.57 | 4.11 | 2.95 | 2.16 | 3.13 | 2.26 |
| GMN w/ Join + Split | 2.63 | 4.16 | 3.01 | 2.31 | 3.01 | 2.34 |
| GMN w/ Change in Length | 2.75 | 4.36 | 3.11 | 2.15 | 3.15 | 2.41 |

## K    MORE DISCUSSIONS ON THE DECOMPOSITION

It is possible to decompose into bigger objects rather than just sticks. As a comparison, we further adopt the hinge-wise decomposition (*i.e.*, decompose the system into particles and hinges), and investigate the performance on both MD17 and Motion Capture. We denote this model as GMN-LH, where H stands for hinges. The results are depicted in Table 13 and Table 14. On MD17, GMN-LH yields a little bit worse performance than GMN-L on several molecules, while giving desirable results on Ethanol and Benzene. On Motion Capture, GMN-LH outperforms GMN-L by a small gap, while is still worse than GMN. By default, we still encourage to perform the decomposition via sticks as sticks are actually the basic building blocks of hinges and other larger rigid objects.

Table 13: Prediction error ($\times 10^{-2}$) on MD17 dataset. Results averaged across 3 runs.

|  | Aspirin | Benzene | Ethanol | Malonaldehyde | Naphthalene | Salicylic | Toluene | Uracil |
| --- | --- | --- | --- | --- | --- | --- | --- | --- |
| EGNN | $14.41_{\pm 0.15}$ | $62.40_{\pm 0.53}$ | $4.64_{\pm 0.01}$ | $13.64_{\pm 0.01}$ | $0.47_{\pm 0.02}$ | $1.02_{\pm 0.02}$ | $11.78_{\pm 0.07}$ | $0.64_{\pm 0.01}$ |
| GMN | $10.14_{\pm 0.03}$ | $\mathbf{48.12}_{\pm 0.40}$ | $4.83_{\pm 0.01}$ | $13.11_{\pm 0.03}$ | $\mathbf{0.40}_{\pm 0.01}$ | $0.91_{\pm 0.01}$ | $\mathbf{10.22}_{\pm 0.08}$ | $\mathbf{0.59}_{\pm 0.01}$ |
| GMN-L | $\mathbf{9.76}_{\pm 0.11}$ | $54.17_{\pm 0.69}$ | $4.63_{\pm 0.01}$ | $\mathbf{12.82}_{\pm 0.03}$ | $0.41_{\pm 0.01}$ | $\mathbf{0.88}_{\pm 0.01}$ | $10.45_{\pm 0.04}$ | $\mathbf{0.59}_{\pm 0.01}$ |
| GMN-LH | $10.25_{\pm 0.06}$ | $52.02_{\pm 0.97}$ | $\mathbf{4.62}_{\pm 0.01}$ | $12.83_{\pm 0.03}$ | $0.41_{\pm 0.01}$ | $1.03_{\pm 0.01}$ | $10.81_{\pm 0.14}$ | $\mathbf{0.59}_{\pm 0.01}$ |

Table 14: Prediction error ($\times 10^{-2}$) on motion capture. Results averaged across 3 runs.

| EGNN | GMN | GMN-L | GMN-LH |
| --- | --- | --- | --- |
| $59.1_{\pm 2.1}$ | $\mathbf{43.9}_{\pm 1.1}$ | $50.9_{\pm 0.7}$ | $\underline{48.7}_{\pm 1.1}$ |

