# OpenReview forum: "Equivariant Graph Mechanics Networks with Constraints"
_ICLR.cc/2022/Conference — ICLR 2022 Poster_

### Official Review · Reviewer_vbzj · 2021-10-25

**Correctness:** 4
**Technical Novelty And Significance:** 3
**Empirical Novelty And Significance:** 2
**Recommendation:** 6
**Confidence:** 4

**Main Review:**

Strengths:

S1. The GMN model delivers exactly what it promises: exact constraint preservation, with E(n) equivariance.

S2. Claim C2, seems like a good idea in general. In fact, leaving hard constraints aside, it seems like a nice alternative to model in Eq. (7) of the E(n) paper, to be able to directly feed the velocity vector to the message function, and perhaps give more expressive power.

S2. The model shows expected advantages in terms of data efficiency over the baselines.

S3. The model shows expected out of distribution generalization properties.

Weaknesses:

W1. The model requires specifying the forward kinematics function for the constraint (to go from generalized positions and velocities to cartesian positions and velocities) as a differentiable function, which in many cases is of a big ask. Furthermore, the approach for writing this constraints does not seem very compositional, for example two separate analyses (forward functions, sets of generalized coordinates, etc) are used to distinguish the hinge and the stick constraints, when in practice a hinge is pretty much two sticks where one of the particles is shared. Would it not be possible to somehow share some of these mechanisms between the hinge and the stick (e.g. just writing the hinge as two sticks in sequence?).

W2. The work focuses a lot on the specific combination of adding exact constraint preservation to the E(N) equivariant model. I believe in most cases the E(n) equivariant model is often used for domains where hard mechanical constraints are not that common (e.g. molecule property prediction, or force prediction). I think it is great to extend the E(N) model this way, but I am not 100% sure how many practitioners interested on the E(n) model will actually end up using the constrain part.

W3. On the other hand there are other baselines from the literature that have looked more explicitly into mechanical systems with hard constraints, which are not currently referenced in the paper. For example, https://arxiv.org/pdf/1810.01566.pdf models a fluid with a floating rigid box made of 64 particles by making a whole prediction for the canonical coordinates of the whole box, and then using the forward kinematic model of the box to obtain the cartesian positions, with an approach that is very similar to equations (6,7,8,9) in this work, so this should probably be referenced. Another common strategy for learning constraints without having to specify the forward kinematics function is to add noise to the training dataset, and ask the model to correct the noise (mujoco rigid systems: https://arxiv.org/pdf/1806.01242.pdf, particle systems: https://arxiv.org/abs/2002.09405, meshes https://arxiv.org/abs/2010.03409), so the model learns to correct constraint violations, and keep constraint violations low during a long rollout, so this should probably be used as a baseline (giving the length of the sticks as an additional static feature). I believe the approach proposed in this paper would probably still do better than training with noise, but it is also much more complex in terms of assumptions that it makes, so it would be an interesting trade off to show. Note that in these type of domains full rotational equivariance is usually not as important since gravity often breaks the symmetry for at least one of the axes.

Other minor comments/observations:

O1. "first recording data and then inducting formulas". If feel the this sentence is a bit extraneous to the paper contributions, but if you want to make that point, probably worth referencing this: https://arxiv.org/abs/2006.11287

O2. "However, all above approaches have ignored the symmetry in physics". There is a large subset of models that partially implement symmetries by enforcing translation equivariance but not rotation equivariance, which should probably be mentioned and referenced in the related work (e.g. https://openreview.net/pdf?id=B1lDoJSYDH, https://arxiv.org/abs/2002.09405, https://arxiv.org/abs/2010.03409).

O3. "Euclidean transformation (translation or rotation) of the input states". I think "reflection" should be part of that list too, since the paper uses E(n) assumptions for the proofs (otherwise this would involve the SE(n) group, and I think the appendix proof would not hold).

O4. I think equation (2) is technically incorrect, since in the E(n) paper the function used to update the hidden node embedding (phi_h in e.q. 6 from E(n) paper), is applied after the sum aggregation, which is not possible in Eq(2). Similarly, the two outputs in eq (5) are confusing, since the actual functional form of phi for vector and non vector outputs will end up having different functional form (as per the last sentence of section 3.3, which again does not clarify whether rho_w is shared or not between the vector and scalars outputs).

O5. Corollary 1. "assume the entries of Z are drawn independently", actually this is quite a strong assumption, there may be many cases in systems where forces, velocities and relative positions are actually parallel to each other (e.g. an object falling in a uniform field with initial zero velocity), so the rank of the matrix would be 1. So probably worth skipping Corollary 1 and just going to Corollary 2.

O6. "the generalized coordinates include the states of particle 0 denoted as ... R^3  the rotation Euler angles of stick 01 ... R^3 ... and 02 .. R3". Actually, if the particle 0 has only 3 degrees of freedom this means they are point particles (e.g. no solid rotation), so would not this mean then sticks 01 and 02, would only have 2 generalized coordinates each (e.g. latitude and longitude), instead of 3?

O7. "The edge feature is provided by a concatenation of the product of charges" Did the model had any trouble learning to perform this product if instead just giving the charge as a node feature of the inputs?

O8. "EGNN deliver much worse performance when the training size is small". Do you have an intuition of why GMN is more data efficient than EGNN, considering both models are similarly equivariant?

O9. Table 3. "are comparable with the original environment" would be possible to add an extra row with the performance GMN trained on those datasets (e.g. what the test performance would be if the model actually saw data from that distribution during training?).

O10. Table 4. Is there an interpretation of why the shared model works worse for the phi_2 that receives the three inputs than the unshared one, but in the case of phi_2 receiving just one input is the other way around?


Small typos/grammar:
"Considering constraints, by no means, is not easy" weird grammar.
"How the size of the training set influences." --> Data efficiency
"We have introduced an normalization" (typo)



**Summary Of The Paper:**

The authors propose Graph Mechanics Networks (GMN), a E(n) equivariant model that explicitly models systems with rigid constraints, with exact constraint preservation by means of using generalized coordinates. In my opinion the main contributions are as follows:

C1. Establish math for writing models that operate on cartesian coordinates, make predictions in very generalized coordinates and then converted back to cartesian. This is an approach that has been used in previous work, but is usually hidden as an implementation detail.

C2. Extend the E(n) (Satorras et al.) equivariant message function to take and output a variable number of vectors with a proof demonstrating that the approach is flexible enough to actually approximate any E(n) equivariant function. I think this is a novel low level modeling contribution.

C3. Extent the E(n) (Satorras et al.) model to build an E(n) equivariant model for exact constraint preservation. Although a bit niche, this is the first model of that nature.

**Summary Of The Review:**

The paper has a reasonable balance of novel contributions along different axes. I am not sure any one of them is sufficiently novel/interesting to meet the ICLR bar on its own, but put together and following corresponding revisions, I think the could make make a bigger ICLR-worthy contribution. My only concern is that it may be hard to reach the right audience, since I suspect many readers would be interested on each of the contributions individually, but not sure many will be interested on all 3 of them simultaneously. I would perhaps recommend including "E(n)" or "equivariant" in the title of the paper to reach a larger audience.

---

> ### Author Response · Authors · 2021-11-14
> **Response to Reviewer vbzj (Page 1)**
>
> We thank the reviewer for the constructive and detailed comments. The reviewer has provided a nice suggestion by including the term "equivariant" in the title of the paper. We accept that it will better emphasize the contribution of our work, and have revised the title accordingly. The response for each concern has been provided as follows.
>
> > **W1: The model requires specifying the forward kinematics function for the constraint.**
>
> Please see **G2** for how our GMN can be applied to complex systems without the need of deriving the exact kinematics. In addition, it is possible that a hinge can be divided into two sticks. To illustrate, we apply the stick kinematics to predict the states of these two sticks and then pool the results by translating the two sticks to the middle of the two predicted positions of the shared particle. In this way, the translated sticks will intersect at the same point. The results are provided in Table 7 in Appendix. It shows the stick-only model performs worse than the hinge-modeled GMN, since splitting the hinge into two sticks overlooks the kinematics at the connecting point. Yet and still, it performs better than EGNN, again verifying the benefit of our proposed constraint modeling.
>
> > **W2: The significance of adding exact constraint preservation to the E(N) equivariant model.**
>
> Thanks for the comment. We believe adding exact constraint preservation to the E(N) equivariant model is valuable to many applications. Particularly considering the two extra evaluations in **G1**, the results on `MD17` and `CMU Motion Capture` do verify the benefit of involving constraints by GMN. Notice that `MD17` is for molecular dynamics prediction and `CMU Motion Capture` is for human motion tracking, both of which facilitate various applications in, for example, drug discovery and security surveillance. Thus, the potential significance of our proposal is somehow verified.
>
> > **W3: About the missing references.**
>
> Thanks for the comment. We have carefully checked the mentioned papers, and found that they all employ graph-based networks to simulate the interaction between particles. Hence, they are all relevant and should be discussed. Sorry for the missing citation. However, they are still obviously distinct from our work, making our contributions still valuable. We list the differences below:
>
> **1.** The first point, as also raised by the reviewer, is that all these papers have not considered rotation equivariance. Perhaps, rotation equivariance is not so demanded for the tasks in these papers due to the involvement of external gravity (the symmetry is broken at the Z-axis), but for the tasks in our paper, equivariance is indispensable for better generalization (see for example the comparison between GNN which is not equivariant and our method in Fig. 4 (right)). Hence, we have developed a general form for Eq. (5-6) as well as discussed when our formulation universally approximates any equivariant function. Such difference clearly distinguishes our work from DPI-Net (https://arxiv.org/pdf/1810.01566.pdf). In DPI-Net, the BoxBath task does share a similar strategy to us by first predicting the canonical coordinates of the whole box and then using the forward kinematic model to obtain the cartesian positions. However, the passing messages in DPI-Net are scalars other than physics-informed directional vectors (positions, velocities, and accelerations) used in our work.
>
> **2.** Another point, compared to the GraphNet-based methods (https://arxiv.org/pdf/1806.01242.pdf, https://arxiv.org/abs/2002.09405, https://arxiv.org/abs/2010.03409), is that they **never** explicitly involve constraints into the model design, and the nodes of the constructed graphs correspond to only particles or meshes but not rigid bodies. In our paper, the graph we build is a mixture of particles, sticks and/or hinges, which requires both particle-level and object-level message passings, not to mention that both processes are equivariant.
>
> The reviewer suggests adding noise to learn constraint satisfaction. While this is indeed an interesting point, we still prefer our current formulation, which is more data-efficient. The evaluations in **G1** do verify the current setting can achieve desirable performance for various practical systems when using the decomposition trick in **G2**.

---

> > ### Author Response · Authors · 2021-11-14
> > **Response to Reviewer vbzj (Page 2)**
> >
> >
> > For other minor comments, we thank the reviewer for organizing them in order and will address them one by one as follows.
> >
> > > O1. "first recording data and then inducting formulas". I feel this sentence is a bit extraneous to the paper contributions, but if you want to make that point, probably worth referencing this: https://arxiv.org/abs/2006.11287
> >
> > **O1:** Thanks for the suggestion. We have revised this sentence as “deriving algebraic expressions from the limited observed data” by borrowing the notions from the paper (https://arxiv.org/abs/2006.11287).
> >
> > > O2. "However, all above approaches have ignored the symmetry in physics". There is a large subset of models that partially implement symmetries by enforcing translation equivariance but not rotation equivariance, which should probably be mentioned and referenced in the related work (e.g. https://openreview.net/pdf?id=B1lDoJSYDH, https://arxiv.org/abs/2002.09405, https://arxiv.org/abs/2010.03409).
> >
> > **O2:** We have fixed the statement and made it more rigorous. The revised version is "However, all above approaches have ignored the rotation symmetry in physics". The mentioned related papers about translation equivariance have been included in this sentence.
> >
> > > O3. "Euclidean transformation (translation or rotation) of the input states". I think "reflection" should be part of that list too, since the paper uses E(n) assumptions for the proofs (otherwise this would involve the SE(n) group, and I think the appendix proof would not hold).
> >
> > **O3:** Yes, "reflection" should be part of that list. We have fixed this point.
> >
> > > O4. I think equation (2) is technically incorrect, since in the E(n) paper the function used to update the hidden node embedding (phi_h in e.q. 6 from E(n) paper), is applied after the sum aggregation, which is not possible in Eq(2). Similarly, the two outputs in eq (5) are confusing, since the actual functional form of phi for vector and non-vector outputs will end up having different functional forms (as per the last sentence of section 3.3, which again does not clarify whether rho_w is shared or not between the vector and scalars outputs).
> >
> > **O4:** For brevity, we combine the two functions (for the hidden vector and acceleration) into one single function in Eq. (2) (and also Eq. (5)), since they share the same input. To eliminate the confusion pointed out by the reviewer, we have added a footnote to explain the difference between the two functions. Note that the parameter is shared for them, following EGNN.
> >
> > > O5. Corollary 1. "assume the entries of Z are drawn independently", actually this is quite a strong assumption, there may be many cases in systems where forces, velocities, and relative positions are actually parallel to each other (e.g. an object falling in a uniform field with initial zero velocity), so the rank of the matrix would be 1. So probably worth skipping Corollary 1 and just going to Corollary 2.
> >
> > **O5:** We admit the possibility of the violation of the assumption in Corollary 1. However, the theoretical discussions in Corollary 1 (and Theorem 1, etc.) can be conducted in a broader context beyond physics. It means the assumption in Corollary 1 will hold potentially if the inputs are not restricted to be positions, velocities, and accelerations. In this sense, Corollary 1 is still meaningful.
> >
> > > O6. "the generalized coordinates include the states of particle 0 denoted as ... R^3 the rotation Euler angles of stick 01 ... R^3 ... and 02 .. R3". Actually, if the particle 0 has only 3 degrees of freedom this means they are point particles (e.g. no solid rotation), so would not this mean then sticks 01 and 02, would only have 2 generalized coordinates each (e.g. latitude and longitude), instead of 3?
> >
> > **O6:** Thanks for raising this point. It is true that if particle 0 is treated as a point particle, then sticks 01 and 02, would only have 2 generalized coordinates each (e.g., latitude and longitude), instead of 3. Yet, we still encourage to allow the Euler angels to be 3D (even this includes redundancy), since it is more convenient to represent the rotation matrix in the forward kinematics (Eq. (13-14)) via the form of 3D angles. Moreover, in the dataset `CMU Motion Capture` in **G1**, some motion joints are allowed to roll; for these cases, the latitude and longitude coordinates are insufficient to describe the states of the joints that connect to the rolling joint. For the above reasons, we tend to define the Euler angels in the 3D space.

---

> > > ### Author Response · Authors · 2021-11-14
> > > **Response to Reviewer vbzj (Page 3)**
> > >
> > > > O7. "The edge feature is provided by a concatenation of the product of charges" Did the model have any trouble learning to perform this product if instead just giving the charge as a node feature of the inputs?
> > >
> > > **O7:** Thanks. Our setting of edge features straightforwardly follows EGNN. We conjecture the initial purpose is to indicate if the interaction between two particles is attraction or repulsion. Per the reviewer’s suggestion, we have tried the setting by just giving the charge as a node feature of the inputs. The results are provided in Table 8 in Appendix. It is hard to tell which setting is better for EGNN and our GMN, and it is possible that the models can learn the interaction of charges from node features.
> > >
> > > > O8. "EGNN deliver much worse performance when the training size is small". Do you have an intuition of why GMN is more data-efficient than EGNN, considering both models are similarly equivariant?
> > >
> > > **O8:** The crucial benefit of GMN against EGNN, as already mentioned in Related Work, is that the constraints are explicitly encoded in the model and GMN does not require learning the constraints from data, which could imply why GMN is more data-efficient than EGNN.
> > >
> > > > O9. Table 3. "are comparable with the original environment" would be possible to add an extra row with the performance GMN trained on those datasets (e.g. what the test performance would be if the model actually saw data from that distribution during training?).
> > >
> > > **O9:** Indeed the full experimental results are provided in Table 6 in Appendix due to the space limit. Below we summarize the results on this topic. We denote GMN trained on (3,2,1) and tested across different scenarios as GMN-Transfer, and GMN trained and tested on the same dataset as GMN-Original. It is observed that GMN has strong generalization capability, since the transfer performance is very close to the original setting. We add this discussion and the tables to Appendix H.
> > >
> > > | \|Train\| = 500 | (3,2,1) | (2,4,0) | (1,0,3) | Average |
> > > | --------------- | ------- | ------- | ------- | ------- |
> > > | GMN-Transfer    | 2.48    | 2.53    | 3.28    | 2.76    |
> > > | GMN-Original    | 2.48    | 2.34    | 3.21    | 2.68    |
> > >
> > > | \|Train\| = 1500 | (3,2,1) | (2,4,0) | (1,0,3) | Average |
> > > | ---------------- | ------- | ------- | ------- | ------- |
> > > | GMN-Transfer     | 2.10    | 2.18    | 2.65    | 2.31    |
> > > | GMN-Original     | 2.10    | 2.01    | 2.44    | 2.18    |
> > >
> > > > O10. Table 4. Is there an interpretation of why the shared model works worse for the phi_2 that receives the three inputs than the unshared one, but in the case of phi_2 receiving just one input is the other way around?
> > >
> > > **O10:** Thanks for raising this interesting observation. When only considering the force information (namely, phi_2 receiving just one input), the acceleration inference for sticks and hinges are similar to each other, since the aggregated force is computed in a similar way; but when considering the concatenation of the force, position, and velocity (namely, phi_2 receiving three inputs), the acceleration inference will be quite different between sticks and hinges, given that their intrinsic kinematics are distinct. This is possibly why the shared model works better for the unshared one for the former case while it works in the other way for the latter case.
> > >
> > > > Small typos/grammar.
> > >
> > > We have fixed the mentioned typos.
> > >
> > > We hope that the reviewer reconsiders the revised version and asks further questions if our responses are still insufficient.

---

> > > > ### Comment · Reviewer_vbzj · 2021-11-18
> > > > **Reply to rebuttal**
> > > >
> > > > Thank you for the comprehensive reply and the additional experiments.
> > > >
> > > > > The first point, as also raised by the reviewer, is that all these papers have not considered rotation equivariance.
> > > > > Another point, compared to the GraphNet-based methods (https://arxiv.org/pdf/1806.01242.pdf, https://arxiv.org/abs/2002.09405, https://arxiv.org/abs/2010.03409), is that they never explicitly involve constraints into the model design.
> > > > > In DPI-Net, the BoxBath task does share a similar strategy to us by first predicting the canonical coordinates of the whole box and then using the forward kinematic model to obtain the cartesian positions. However, the passing messages in DPI-Net are scalars other than physics-informed directional vectors (positions, velocities, and accelerations) used in our work.
> > > >
> > > > I agree with the authors that none of the papers in the literature have utilized this specific combination of techniques, however, my point was that many of the ideas, had already been used in related research, just not all at once. So that takes away a bit of the novelty since I think technically the only fully novel part is the modification of the E(n) model.
> > > >
> > > > Still, I think thinking of this approach to combine exact constraint preservation with rotationally equivariant models is not trivial. Also unifying this into a model that may be useful for practical applications across several domains is important, and the additional experiments presented on MD17 and MoCap proof this. So for now I am happy to at least maintain my already positive rating, and wait for the rest of the discussion.
> > > >
> > > > I have a few questions about the extra experiments:
> > > >
> > > > Q1. Can the MD17 results on Table 4 considered state of the art? I think I would be clearly willing to raise my score further if this model can be considered to be state of the art by the community on a well known task like MD17. However, I think other papers making spatial predictions on MD17 report error on forces, and it would hard to impose a constraint on the forces, unless you assume an integrator, so I assume you are reporting error in positions, is this correct? Can you clarify if the results from the baselines from Table 4 taken from the literature, or computed by you? My impression from the appendix is that it is the latter, and that the task is actually slighly different to what other models in the literature do. Would it be possible at all to actually report on the same forces task, and be able to not only reproduce baselines from the literature, but actually matching baseline values from the literature in Table 2?
> > > >
> > > > Q2. With respect to **G2** and the partial constraints, I think it is indeed very nice to show that preserving partial constraints, can actually help, because indeed having to provide only partial kinematics will make the approach much more applicable, and it is not immediately obvious that partial inexact constrains is better than no constraints.
> > > >
> > > >
> > > > Q3.
> > > > > We observe that the lengths of
> > > > chemical bonds remain very stable during the simulation,
> > > > making it reasonable to model the bonds as sticks.
> > > >
> > > > > perform worse than others
> > > > on Ethanol, possibly because Ethanol is a relatively small molecule with simple structure, where
> > > > considering the bond constraints possibly makes less benefit but instead hinders the learning
> > > >
> > > > The results that show that performance for Ethanol on MD17 is worse is not too surprising, since indeed the bonds distances in MD17 do not have to be exactly preserved. It would be nice to show the standard deviation in bond distance for each molecule in MD17, as if I understand correctly, this would give an lower bound on the best error can be achieve by this model. I really like the attempt to apply this to MD17, but if this is an issue this means this model may be a quick win in the short term, but may not be really useful in the long term.

---

> > > > > ### Author Response · Authors · 2021-11-19
> > > > > **Further Responses to Reviewer vbzj**
> > > > >
> > > > > Thank you for your recognition of our contributions particularly given the additional experiments presented on MD17 and MoCap. We are willing to address your further questions below.
> > > > >
> > > > > > ``Q1. Can the MD17 results on Table 4 considered state of the art? I think I would be clearly willing to raise my score further if this model can be considered to be state of the art by the community on a well known task like MD17. However, I think other papers making spatial predictions on MD17 report error on forces, and it would hard to impose a constraint on the forces, unless you assume an integrator, so I assume you are reporting error in positions, is this correct? Can you clarify if the results from the baselines from Table 4 taken from the literature, or computed by you? My impression from the appendix is that it is the latter, and that the task is actually slighly different to what other models in the literature do. Would it be possible at all to actually report on the same forces task, and be able to not only reproduce baselines from the literature, but actually matching baseline values from the literature in Table 2?``
> > > > >
> > > > > Thanks for your question. Yes, as you said, the experiments here report errors on positions other than forces. The reason why we choose this setting is that our paper is exploring how involving the **geometrical constraint** (not force constraint) can regularize the prediction of positions. As also pointed out by you, it would be hard to impose a constraint on forces, unless some other techniques are used, which will probably be beyond the main scope of this paper.
> > > > >
> > > > > Since it is the first time that MD17 is used for position prediction, there is no state of the art previously and we have to implement all compared methods by using their official codebases. This actually helps in maintaining experimental fairness. For example, for the parts (such as the interaction force function, $\phi_1$ in Eq.(5)) overlapped with EGNN, we keep them exactly the same in GMN (in terms of network structure, hyper-parameters, etc.); in this way, our contributions upon EGNN are sufficiently and convincingly verified.
> > > > >
> > > > > We understand the concern of the reviewer. But we also believe that it is indeed meaningful to set up a new evaluation protocol for a new problem, if the problem itself is valuable and the reported results are fairly compared.
> > > > > We have released all our code sources (including the implementation of all baselines) to facilitate the following studies on this new evaluation protocol and really hope the reported results would become a new benchmark on MD17.
> > > > >
> > > > > > ``Q2. With respect to G2 and the partial constraints, I think it is indeed very nice to show that preserving partial constraints, can actually help, because indeed having to provide only partial kinematics will make the approach much more applicable, and it is not immediately obvious that partial inexact constrains is better than no constraints.``
> > > > >
> > > > > Thanks for this positive comment. We are also happy to see that preserving partial constraints actually helps in MD17 and Motion Capture.
> > > > >
> > > > > > ``Q3. The results that show that performance for Ethanol on MD17 is worse is not too surprising, since indeed the bonds distances in MD17 do not have to be exactly preserved. It would be nice to show the standard deviation in bond distance for each molecule in MD17, as if I understand correctly, this would give an lower bound on the best error can be achieve by this model. I really like the attempt to apply this to MD17, but if this is an issue this means this model may be a quick win in the short term, but may not be really useful in the long term.``
> > > > >
> > > > > Thank you for your suggestion! We agree that the bonds distances in MD17 do not have to be exactly preserved. We thus measure the standard deviation of the bond distances of all input and target frames for each molecular, and report the metric below:
> > > > >
> > > > > |                            | Aspirin | Benzene | Ethanol | Malonaldehyde | Naphthalene | Salicylic | Toluene | Uracil |
> > > > > | -------------------------- | ------- | ------- | ------- | ------------- | ----------- | --------- | ------- | ------ |
> > > > > | std               | 0.030   | 0.020   | 0.039   |0.043        | 0.028      | 0.029     | 0.031   | 0.029  |
> > > > >
> > > > > We find that the std is very small (more than 10 times smaller) compared with the prediction errors in Table 4, which implies that the "lower bound" could not be the obstacle for applying GMN in molecular data like MD17. In addition, in order to deal with constraints in a more flexible manner, we propose to replace the hand-crafted forward kinematics with a black-box learnable equivariant function, as requested by Reviewer #1 (4PY9) and #3 (Dcng). We denote this method as GMN-L.
> > > > > GMN-L interestingly achieves very promising performance for all molecules including Ethanol. We would like to refer the reviewer to Table 4 in the revised paper for the detailed results.
> > > > > We are very willing to explain more if there is still any concern.

---

> ### Author Response · Authors · 2021-11-17
> **Welcome to comment**
>
> Dear reviewer,
>
> We have clarified how our method can be applied to complex systems and provided the supporting experiments on two real-world applications. The discussions about the comparison with the mentioned papers are also contained in the revised paper. Please feel free to ask any questions.
>
> Thanks.

---

### Official Review · Reviewer_Dcng · 2021-11-01

**Correctness:** 4
**Technical Novelty And Significance:** 3
**Empirical Novelty And Significance:** 3
**Recommendation:** 8
**Confidence:** 4

**Main Review:**

Strengths:

The paper is decently clearly written (modulo some typos/phrasings noted below, but nothing that impedes the understanding of the reader), provides all necessary proofs and makes 2 main contributions

- formulating an algorithm which incorporates Generalized Coordinates and inverse/forward kinematics
- proposes an extension of the EGNN framework from working on single vectors to matrices, including some proofs about the equivariance and expressivity.


Incorporating the constraints clearly helps generalization on  the experiments

Weaknesses:

- My main criticism is with regards to lack of novelty/generality: one has to manually derive the forward/inverse kinematics of the constrained system, which requires knowledge of the underlying physics already...at which point one might want to go one step further and derive some approximation as well? Alternatively, I would have expected the incorporation of the generalized coordinates to be done in a more black box fashion, i.e. encoding the knowledge that there are *some* generalized coordinates in the architecture and letting the models derive *which* we are talking about (this could possibly be attempted with invertible neural networks for the kinematics and some form of constraint-seeking algorithm that tries to minimize the total number of dimensions used in the generalized coordinate latent for example?)
- only studied on a single setting, but this is a minor nitpick as it's similar to other previous works
- reporting the constraint violation in the main body is only marginally meaningful as of course a system which enforces the constraints *by design* will achieve 0 constraint violation and that part in particular is not learned.

Minor nitpick:
- the paper mentions rotation and translation in the intro but then focuses mainly on rotation? doesn't change much for the final results but it confused me a bit


Typos/phrasings  (not accounted for in my review, just as feedback for the authors):

- page 2: "have achieved desired performance on the N-body system...lacks of constraint" is meant to  say "have achieved desirable performance ....lack constraints"?
- page 3 "is proportion to the its" "is proportional to the" ?
- page 4: "despite the desired performance" => "desireable" again?
- page 5: "can be generalized to the function with multiple input vectors" => to functions with multiple input vectors?

**Summary Of The Paper:**

The paper focuses on the formulation of a graph neural network approach towards the learning of update rules of constrained systems, extending prior work (Interaction entwork, EGNN) by learning the updates of constrained components in the generalized coordinates expressing those constraints and incorporating forward and inverse kinematics in the algorithm.
Comparisons are made with Linear, Basic GNN, TensorForce Networks, SE3-Transformers, Radial Field and EGNNs by evaluating them on a constrained version of the N-body problem introduced by Kipf et al. 2018 and it is shown that the proposed method better adheres to constraints and generalizes better on the studied problem.

**Summary Of The Review:**

The paper presents and incremental improvement over previous works and was done cleanly, so a marginal accept. If it had been exploring a fully learned generalized coordinate transformation that uses an inductive bias but does not need to hand design the coordinates and kinematics I'd give a higher score.

://EDIT: updated my score

---

> ### Author Response · Authors · 2021-11-14
> **Response to Reviewer Dcng**
>
> We are thankful to the reviewer for the constructive comments. We address the reviewer’s concerns as follows.
>
> > **Q1: The concerns on the novelty/generality.**
>
> Please see **G2** for how GMN can be applied to complex systems, without the need to hand design the coordinates and kinematics of the entire system. The reviewer has mentioned the incorporation of the generalized coordinates to be done in a more black-box fashion. Yet, we would like to highlight that involving the domain knowledge of the rigid constraint into GMN is still what we pursue in the first place, and it requires fewer training samples and ensures more data efficiency compared to the data-driven counterparts. Specifically, as we explained in **G2**, using the current hand-crafted form of Eq. (9) is still capable of handling complex systems. Still, we really think that the reviewer has suggested an interesting idea, and will explore it in future work.
>
>
> > **Q2: Only studied in a single setting.**
>
> This is no longer an issue. We have provided two additional evaluations on molecular dynamics prediction and motion capture. The results generally support the advantage of our GMN compared to other compared methods. Please refer to **G1** for more details.
>
> > **Q3: Reporting the constraint violation in the main body.**
>
> Thanks for the reminder. Our initial purpose of showing constraint violation is to verify the property of constraint satisfaction by GMN. We agree with the reviewer that this part is not sufficiently informative, and have moved the table of the constraint violation to Appendix.
>
>
> > **Q4: Other minor comments.**
>
> GMN is equivariant to both rotations and translations. Since the translation equivariance is naturally satisfied by using relative positions in Eq. (5), we only need to further enforce rotation equivariance. We have mentioned this in the footnote on Page 5. Besides, we have corrected the mentioned typos/phrasings. Thanks for the reminder.

---

> > ### Comment · Reviewer_Dcng · 2021-11-14
> > **I'll increase my score if you can include some black box baseline**
> >
> > Thank you, you have reduced my criticisms of the work to the requirement of domain knowledge. Since I already gave a marginal accept I am reluctant to improve my score much further without addressing this concern on some fundamental level so in the spirit of giving constructive criticism I will say that barring influences from the discussion with other reviewers I'm willing to raise my score if there are ablations added that show what happens if you mis or underspecify constraints (e.g.,  can you specify only the distance, but not the angle as a hard constraint, and vice versa? (I think this is a bit different than your added experiment) Is the model still able to learn to some degree of given the wrong constraints? ) and whether learning the forward and backward kinematics is possible to learn with a naive blackbox (e.g. the additive coupling invertible layers present in the memcnn baseline, or simply using two separate learnable projection layers). If the former ablations are favourable it shows robustness of the work against slight error in the domain expertise (staying useful in very complicated and possibly nuanced beyond human understanding domains) while the latter would give some evidence that hardcoding the constraints replaces a nontrivial amount of learning complexity.
> >
> > Lacking this or if th discussion with other reviewers convinces me otherwise I'll retain my score, otherwise I'd be willing to upgrade to 7. I hope the authors think of this as a fair and constructive critique?

---

> > > ### Author Response · Authors · 2021-11-15
> > > **The black box baseline has been included (Part 1)**
> > >
> > > Thank you for these nice suggestions! We agree that these are very interesting points to explore. We separate your questions into two points, and conduct supportive experiments to address your concerns, respectively. We have contained the results of the requested black-box baseline in Table 3 in the main body of the revised paper.
> > >
> > > Please feel free to ask any further questions if our responses are still insufficient.
> > >
> > >
> > > > ``What happens if you mis or underspecify constraints, e.g., can you specify only the distance, but not the angle as a hard constraint, and vice versa?``
> > >
> > > The reviewer has raised an interesting question that would serve as a test of the robustness of our model w.r.t. the noisy constraints.  This scenario would sometimes arise in real-world scenarios where we might not be certain about the exact connectivity of the rigid body, and thus would involve slight errors in domain expertise. Since our paper focuses on the distance constraint other than the angle constraint, the following investigations will only involve noise into the stick connectivity. We design three random perturbation operations on the input rigid body prior: #1 (Join). randomly selecting 2 isolated particles and joining them as if there is a stick connecting; #2 (Split). randomly selecting an existing stick and splitting it as two isolated particles; and #3 (Join + Split). conducting operation #1 and #2 at the same time. Note that the operation is conducted independently for every training sample each time it is fed into the network.
> > >
> > > We summarize the results in the table below (Table 12 in Appendix J of the revised paper).
> > >
> > >
> > >
> > > | \|Train\| = 500   | (3,2,1) | (5,3,3) | (8,6,0) |
> > > | ----------------- | ------- | ------- | ------- |
> > > | GMN               | 2.48    | 4.08    | 2.84    |
> > > | GMN w Join        | 2.59    | 4.27    | 2.98    |
> > > | GMN w  Split      | 2.57    | 4.11    | 2.95    |
> > > | GMN w  Join+Split | 2.63    | 4.16    | 3.01    |
> > >
> > >
> > >
> > > | \|Train\| = 1500  | (3,2,1) | (5,3,3) | (8,6,0) |
> > > | ----------------- | ------- | ------- | ------- |
> > > | GMN               | 2.10    | 2.86    | 2.22    |
> > > | GMN w Join        | 2.22    | 3.16    | 2.37    |
> > > | GMN w  Split      | 2.16    | 3.13    | 2.26    |
> > > | GMN w  Join+Split | 2.31    | 3.01    | 2.34    |
> > >
> > > We observe that these perturbations, although somehow hinder the performance, in general do not jeopardize the performance too much (difference in MSE $\leq$ 0.30), indicating that GMN is not sensitive to slight errors of the input physical prior of the constraints and it is still able to learn to some degree of given the wrong constraints.

---

> > > > ### Author Response · Authors · 2021-11-15
> > > > **The black box baseline has been included (Part 2)**
> > > >
> > > >
> > > > ### Learnable FK
> > > >
> > > > > ``Whether learning the forward and backward kinematics is possible to learn with a naive blackbox``
> > > >
> > > > It is indeed instrumental to discuss whether a learnable black-box function, which requires less domain knowledge, could also yield competitive performance, and if our hand-crafted FK still shows advantage over the learnable counterpart. To answer these questions, we replace the hand-crafted part (Eq. (7-9) as well as the Euler angle computations in Sec. 3.2) with the following equations:
> > > >
> > > > $v_i^l = \phi (h_i^{l-1})v_i^{l-1}$ + $ \rho (q'', x_{ki}^{l-1}, f_i^l),x_i^l = x_i^{l-1} + v_i^l$ where $q''=\ddot{q}_{k}^l$
> > > > and $\rho$ is the equivariant message passing layer we propose in Sec. 3.3. By this design, the parameterized FK preserves its equivariant property (and the theoretical universality), and compared to EGNN, it additionally leverages the information from the object-level generalized coordinates $\ddot{q}_k^l$. We denote this variant of GMN as GMN-L. Moreover, the parameterized FK inevitably loses the constraint-preserving property compared with the exact FK, therefore we also augment it with explicit constraint regularization, akin to what we did to EGNNReg. We hence denote this variant as GMN-LReg.
> > > >
> > > > We evaluate the performance of GMN-L, GMN-LReg and compare them with GMN with exact FK as well as EGNN and EGNNReg in the table below (see Table 11 in Appendix I of the revised paper).
> > > >
> > > >
> > > >
> > > >
> > > > | \|Train\| = 500 | (1,2,0)  | (2,0,1)  | (3,2,1)  | (0,10,0) | (5,3,3)  |
> > > > | --------------- | -------- | -------- | -------- | -------- | -------- |
> > > > | EGNN            | 2.81     | 2.27     | 4.67     | 4.75     | 4.59     |
> > > > | EGNNReg         | 2.94     | 2.66     | 7.01     | 5.03     | 6.31     |
> > > > | GMN-L           | **2.32** | **2.09** | **3.19** | 3.88     | 4.34     |
> > > > | GMN-LReg        | 2.52     | 2.23     | 3.34     | **3.67** | **4.31** |
> > > > | GMN             | 1.84 | 2.02 | 2.48 | 2.92 | 4.08 |
> > > >
> > > >
> > > >
> > > > | \|Train\| = 1500 | (1,2,0)  | (2,0,1)  | (3,2,1)  | (0,10,0) | (5,3,3)  |
> > > > | ---------------- | -------- | -------- | -------- | -------- | -------- |
> > > > | EGNN             | 2.59     | 1.86     | 2.54     | 2.79     | 3.25     |
> > > > | EGNNReg          | 2.74     | 1.58     | 2.62     | 3.03     | 3.07     |
> > > > | GMN-L            | 1.93     | **1.56** | **2.28** | 2.72     | 3.03     |
> > > > | GMN-LReg         | **1.91** | 1.88     | 2.49     | **2.61** | **3.00** |
> > > > | GMN              | 1.68 | 1.47 | 2.10 | 2.32 | 2.86 |
> > > >
> > > > We interestingly find that GMN-L consistently outperforms EGNN in various settings (as well as the regularized version), which again verifies both the validity of our proposed equivariant message passing layer and the efficacy of leveraging object-level message (i.e., $\ddot{q}_k^l$) for the inference of FK.
> > > > At the same time, GMN-L and GMN-LReg yield a minor gap with GMN, showing the evidence that hardcoding the constraints replaces a nontrivial amount of learning complexity. We sincerely thank the reviewer again for motivating us to design the learnable FK, which better strengthens our contribution of the equivariant message-passing layer and simultaneously addresses the concerns raised by other reviewers.

---

> > > > > ### Comment · Reviewer_Dcng · 2021-11-15
> > > > > **Thank you very much, some clarifications**
> > > > >
> > > > > Thank you very much, this is indeed very interesting to see. To clarify
> > > > >
> > > > > - in the noisy constraint experiment, you add noise to the distance section of the problem,since you do not constrain the angle already, yielding to the hard constraints specifying requirements that are not there, or emitting constraints that are there? If I understood this correctly, did you also run experiments where systematically add an error to a constraint?
> > > > > - in the forward kinematics constraint, you fully remove the need for specifying the coordinate system?

---

> > > > > > ### Author Response · Authors · 2021-11-16
> > > > > > **More clarifications on the noisy constraint experiment and the forward kinematics constraint**
> > > > > >
> > > > > > Thank you for your interest! We answer your further questions below.
> > > > > >
> > > > > > > in the noisy constraint experiment, you add noise to the distance section of the problem,since you do not constrain the angle already, yielding to the hard constraints specifying requirements that are not there, or emitting constraints that are there? If I understood this correctly, did you also run experiments where systematically add an error to a constraint?
> > > > > >
> > > > > > In the extra experiments above, as we specified, we added three perturbations on the input constraints, namely 1. randomly adding a constraint, 2. randomly removing a constraint, and 3. randomly adding and removing a constraint. Therefore, our experiments above did include both ``specifying requirements that are not there`` and ``emitting constraints that are there``. We see that the negative effect these perturbations exert on our model is very limited.
> > > > > >
> > > > > > To further clarify this point ``systematically adding an error to a constraint``, we now consider an extra operation: randomly adding noise to the length of sticks. By this means, we add a Gaussian noise $\mathcal{N}(0, 0.1L)$ to the length of a random stick each time the training sample fed into the network, where $L$ is the original length of the selected stick. This is another perspective of adding systematic error to a constraint, apart from what we did last time. We denote this operation as Change in length, and show the performance of our model in the table below (this new operation is highlighted in boldface). We also update the results in Table 12 of Appendix J.
> > > > > >
> > > > > >
> > > > > >
> > > > > > | \|Train\| = 500        | (3,2,1) | (5,3,3) | (8,6,0) |
> > > > > > | ---------------------- | ------- | ------- | ------- |
> > > > > > | GMN                    | 2.48    | 4.08    | 2.84    |
> > > > > > | GMN w Join             | 2.59    | 4.27    | 2.98    |
> > > > > > | GMN w  Split           | 2.57    | 4.11    | 2.95    |
> > > > > > | GMN w  Join+Split      | 2.63    | 4.16    | 3.01    |
> > > > > > | **GMN w Change in length** | 2.75    | 4.36    | 3.11    |
> > > > > >
> > > > > >
> > > > > >
> > > > > >
> > > > > >
> > > > > > | \|Train\| = 1500       | (3,2,1) | (5,3,3) | (8,6,0) |
> > > > > > | ---------------------- | ------- | ------- | ------- |
> > > > > > | GMN                    | 2.10    | 2.86    | 2.22    |
> > > > > > | GMN w Join             | 2.22    | 3.16    | 2.37    |
> > > > > > | GMN w  Split           | 2.16    | 3.13    | 2.26    |
> > > > > > | GMN w  Join+Split      | 2.31    | 3.01    | 2.34    |
> > > > > > | **GMN w Change in length** | 2.15    | 3.15    | 2.41    |
> > > > > >
> > > > > > As we observe, the performance with noise injection still remains close to the unperturbed scenario, indicating GMN is not sensitive to slight changes in length. In summary, we design two views of injecting systematic error to the input constraints by 1. Joining or splitting the stick and 2. Adding noise to the length of the stick. We hope our setups well clarify your concerns.
> > > > > >
> > > > > >
> > > > > >
> > > > > > > in the forward kinematics constraint, you fully remove the need for specifying the coordinate system?
> > > > > >
> > > > > > Yes. There is no need to specify the coordinate system in the learned-FK we design. The equivariant layer $\rho$ is responsible for decoding the object-level message $\ddot{q}_k^l$ to particle-level dynamics, both of which are black-box, and do not require computing via a coordinate system.
> > > > > >
> > > > > > We are happy to further clarify if anything is still not clear.

---

> ### Author Response · Authors · 2021-11-17
> **Thanks for your feedbacks**
>
> Dear reviewer,
>
> Thank you very much for your further feedbacks. For your last requirement of the clarifications on the noisy constraint and learnable forward kinematics, we have provided more clarifications as well as the supporting experiments. Please feel free to ask any further questions.
>
> Thanks.

---

> ### Author Response · Authors · 2021-11-19
> **The black-box baseline is evaluated on the real-world dataset as well**
>
> Dear reviewer,
>
> In our last response, we evaluated the performance of the black-box baseline (GMN-L) on the simulated data. Now, we have further contrasted the performance of GMN-L against other methods on the real-world datasets: MD17 and Motion Capture. The results are provided in Table 4 and 5 in the revised paper. It is interesting that GMN-L still performs promisingly.
>
> Please feel free to ask any questions regarding our new results.
>
> Thanks.

---

### Official Review · Reviewer_F8Jp · 2021-11-01

**Correctness:** 4
**Technical Novelty And Significance:** 1
**Empirical Novelty And Significance:** 1
**Recommendation:** 5
**Confidence:** 2

**Main Review:**

**Strengths**

The presentation is clear and professional.

The task is well defined and has some physical relevance.

The supplementary materials are extensive and include a quality code repository that is well documented.

**Weaknesses**

There is a lack of novelty and scope in the work. The following problems are fundamental to the contribution and it is not clear to me how the authors could remedy these faults within the rebuttal period unless they would like to flatly refute the points:

1. Most of the core ideas presented here are captured and applied more widely, including to real-world scenarios such as the case of robotics mentioned in the introduction of this work, in ​​Deep Lagrangian Networks: Using Physics as Model Prior for Deep Learning Lutter et al. ICLR 2019. (This reference is missing.)

2. The scope of the theoretical and empirical contributions of the paper are limited to very small systems of sticks and hinges. There are existing works in this area that scale to more complicated rigid systems without the need for hard coded physics (Sanchez-Gonzalez et al. ICML 2018), to large-scale systems of thousands of particles (Sanchez-Gonzalez et al. ICML 2020), and to systems with constraints (Pfaff ICLR 2021). The contributions are thus quite limited in the context of the achievements of the field (and in particular when the work of Lutter et al. is considered.)

3. The general formulation of an equivariant message passing layer given in Equation 11 does appear novel, although the actual use case with $Z = (f,x,v) = (ma, x, v)$ is very close to the extension given by ​​Satorras et al. (EGNN) in their equation 7 that shows how to extend their method to using velocities. It is trivial to extend their method to include an equivariant treatment of accelerations ($v^{l+1}_i = v^{l}_i + a^{l+1}_i$). It is also in the spirit of the work presented here to include such a physical inductive bias (i.e. that the rate of change of the velocity $v$ is set to be exactly the acceleration $a$ and so on).

All the tables should include errors. I appreciate that the results suggest that the GMN is the dominant model in this system but it is still necessary to indicate the variation in the recorded performances.

It is also not correct to state that ‘GMN meets constraints spontaneously.’ The constraint satisfaction has been hand crafted into the model whereas the compared models are required to learn the constraints. This is nearly a minor point but it does make an incorrect claim.

The self-containment of the paper, as described in Appendix B with reference to the derivation of the dynamics of sticks and hinges, whilst admirable, is not relevant to the novelty or quality of the scientific contribution of the work.

**Minor comments**

The following notes are very minor and may be subjective and as such I have not factored them in to my score for the paper. I would not update the score in either direction if the authors ignore or act on these notes, they're just things that I spotted as I went through the paper.
- It's unclear to me what the first sentence of the abstract is trying to say. 'Prevailing yet challenging' is a little confusing to me, is it not the case that challenging topics are also prevalent?
- 'mainly stem from that...' in the second sentence of the abstract needs to be rephrased
- 'just like what physicists used to do' in the introduction is a little childish, though perhaps we should wonder what it is that physicists are now getting up to
- 'Coulombian force' is an unusual variation on 'Coulomb force'
- Page 4 'to inference the acceleration' should be 'to infer'


### Rebuttal Update
I have raised my score (3->5) in light of the additional experiments and lowered my confidence (3->2) as I have not been able to engage as thoroughly as the other reviewers who are apparently satisfied with the paper.

**Summary Of The Paper:**

This paper develops a model that augmented interaction networks with mechanical constraints and develops equivariant message passing schemes to ensure physical realism in the model outputs. The work focuses on a set of toy systems that contains particles connected by sticks and hinges and the developments are made for these systems in particular. There are some results on generalised equivariant message passing and the universality of the proposed method.

**Summary Of The Review:**

The ideas presented are physically well motivated but only incrementally advance on the state of the field by combining many existing ideas, and the application scope is very narrow so it is not the case that the paper derives a significant contribution by showing that these techniques taken together are greater than the sum of their parts etc. The presentation is very professional and their is some novelty in extending the equivariant message passing scheme of Satorras et al., but these are not strong enough contributions to justify acceptance alone.

---

> ### Author Response · Authors · 2021-11-14
> **Response to Reviewer F8Jp (Page 1)**
>
> We thank the reviewer for the provided comments. After reading the comments seriously, we are afraid that the reviewer has probably misunderstood the contributions and the novelty of our paper. We will try our best to eliminate the misunderstandings via the following responses, and sincerely hope that the reviewer raises any further questions if he/she is still confused by our answers.
>
> > **Q1: On the novelty.**
>
> We would like to emphasize that our contributions do **NOT** lie in the combinations of existing ideas. We emphasize the novelty and significance of our paper from the following aspects:
>
>
> 1.    Our goal of simulating the dynamics of the constrained N-body system is novel and has never been explored before as far as we know. We thank the reviewer for raising the comparison with the work DeLaN (Lutter et al., 2019). However, DeLaN mainly focuses on the (forward or inverse) dynamics prediction on a single rigid object (the robot), whereas our paper is concerned with the prediction of multiple interacting rigid and structural objects.
>
> 2.    This different formulation from DeLaN requires us to develop different techniques to address the corresponding challenges. In particular, we propose a novel pipeline that is mainly composed of the three stages: representing the inter-object interactions in the Cartesian space via graphs, mapping the force messages from the Cartesian space to the generalized space, and computing the updated Cartesian coordinates via forward kinematics. As a rough comparison, DeLaN is only applicable for the third stage, exhibiting a smaller scope than our paper.
>
> 3. Moreover, DeLaN does not consider equivariance, probably because the configuration of the robot can be uniquely described by the local and relative coordinates (usually the joint angles), where equivariance is not so demanded. In our task, the states of and interactions among multiple objects are described by Cartesian coordinates, which demands the model we built to be equivariant w.r.t. rotations and translations. However, it is nontrivial to derive equivariant functions. We have derived a more general form than EGNN (Satorras et al., 2021) with necessary theoretical guarantee.
>
> 4. We have provided additional evaluations on two real-world applications which contain complex systems including molecules and motion graphs, as discussed in **G1**. Our GMN generally outperforms all compared methods, which demonstrates the potential application of our proposed idea and method for various real-world tasks.
>
> > **Q2: The scope of the theoretical and empirical contributions of the paper.**
>
> Thanks for the comment. Yet, our contributions are NOT limited to small systems of sticks and hinges. As already explained in **G1**, we have shown how the simple particle/stick models can be used to characterize real and complex systems from molecular dynamics prediction to motion capture, which are central to various applications, such as drug discovery and security surveillance. We hope the reviewer will reconsider the significance of our contributions, particularly given the extra evaluations on the two real-world applications.
>
>
> The reviewer has mentioned the references for physics simulation by using GraphNets (Sanchez-Gonzalez et al.), a specific form of GNN. While these works do employ graphs consisting of particles/meshes to model complicated rigid systems, they lack consideration in several aspects compared to our work. The first point is that they are not orthogonality-equivariant. For the tasks in our paper, equivariance is indispensable for better generalization. Actually, our experiments have exhibited that GMN and other equivariant GNN methods outperform GNN significantly, which somehow (although not exactly) validates the advantage of GMN over these GraphNet-based methods since they are just specific cases of GNN if we omit their exact implementation. Another point is that they never explicitly involve constraints in the model design, and the nodes in the constructed graphs correspond to only particles/meshes but not rigid bodies. In our paper, the graph we build is a mixture of particles, sticks and/or hinges, which requires both particle-level and object-level message passing, not to mention that both processes are equivariant. We have cited and discussed these references in our paper.

---

> > ### Author Response · Authors · 2021-11-14
> > **Response to Reviewer F8Jp (Page 2)**
> >
> >
> > > **Q3: The general formulation of the equivariant message passing layer.**
> >
> > We are not sure if we capture the reviewer’s idea on this question; if not, please remind us. To eliminate confusion, we denote the equation numbers in the EGNN paper by further adding the superscript * to distinguish them from the equations in our paper.
> >
> > It is likely that the reviewer has misunderstood why we use $Z=(ma,x,v)$ in Eq. (11). Eq. (11) (i.e. $\varphi(Z,h)$) is a general form of Eq. (10) (i.e. $\varphi_{\text{egnn}}(x,h)$). Our Theorem 1 and its two corollaries state that extending the input $x$ to $Z$ can probably increase the universality of the equivariant function. Yet, this extension is completely different from extending Eq. (4)* to using velocities by Eq. (7)* in EGNN (Satorras et al.).
> > From Eq. (4)* to Eq. (7)* , the authors of EGNN want to show how to extend the dynamics update from positions to velocities, and this extension can of course be applicable to accelerations. However, this kind of extension has nothing to do with the universality, since both Eq. (4)* and Eq. (7)* apply the same form of the equivariant message (i.e., $\sum_{j\neq i} (x_i^l-x_j^l)\phi_x(m_{ij})$) which is indeed equivalent to Eq. (10) in our paper. Only when we replace Eq. (10) with Eq. (11), we can improve the expressivity for the update of either position in Eq. (4)* or velocities in Eq. (7)* in the EGNN paper. Yet, EGNN only applies Eq. (10), while our paper derives Eq. (11) and proves its better expressivity than Eq. (10) in theory. This is where our theoretical novelty lies.
> >
> >
> > > The self-containment of the paper, as described in Appendix B with reference to the derivation of the dynamics of sticks and hinges, whilst admirable, is not relevant to the novelty or quality of the scientific contribution of the work.
> >
> > We thank the reviewer for recognizing our derivations in Appendix B. Although this is not the key point of our contributions, it is indeed relevant to the novelty or quality of the scientific contribution of the work. With these derivations, we can implement a new simulated dataset that generalizes the N-body system to the constrained version. We also release our code of data generation to facilitate the following studies in related domains.
> >
> > > **Other minor comments.**
> >
> > We have added the standard deviations for all methods over 3 random runs in Table 1. We thank the reviewer for providing the suggestions on our presentations, and we have enhanced the raised phrases and sentences in the revised version.

---

> > ### Comment · Reviewer_F8Jp · 2021-11-22
> > **Further comments**
> >
> > Firstly, my apologies for the delayed response.
> >
> > #### **Re Q1**
> >
> > 1. Simulating the dynamics of constrained mechanical systems is a well explored topic. Charitably, I will assume the authors have made a typo here and mean something like 'learning to ... using neural networks' because there are textbooks written on the subject of simulating constrained mechanical systems. On learning to simulate constrained mechanical systems using neural networks, the reference I provided in the original review (Sanchez-Gonzalez et al. ICML 2018) is on the topic of simulating the mechanics of constrained many-body systems.
> >
> > 2. and 3. I appreciate that this paper combines other ideas beyond those found in DeLaN, but the authors should acknowledge that the DeLaN paper does cover some of the ideas that are presented in this work. As the authors state, the scope may be smaller but the content does overlap.
> >
> > 4. (see below)
> >
> > #### **Re Q2**
> >
> > Thank you for these **additional** experiments, I will revise my score upwards in light of this demonstration.
> >
> > #### **Re Q3**
> >
> > I understand what has been done in this work, my point is that it is not a great extension of the work in EGNN. I don't think the response here adds to the presentation given in the paper.
> >
> > #### **Re the self-containedness**
> >
> > There seems to have been some confusion here, I state that self-containment is **not** relevant to the novelty or quality of the scientific contribution of the work.
> >
> > Overall, given the other reviewers apparent satisfaction and the efforts in the additional experiments I will raise my score. I will also lower my confidence in light of my delayed response to the rebuttal, and apologise again to the authors for not being more engaged during the rebuttal period.

---

> > > ### Author Response · Authors · 2021-11-23
> > > **Thanks for your further comments**
> > >
> > > Dear Reviewer,
> > >
> > > We really appreciate that you have recognized the efforts that we put into the extra experiments and increased your score. We are also thankful for your reference to DeLaN and the work by Sanchez-Gonzalez et al.  (ICML 2018). If necessary, we would like to emphasize the main difference again: our work has further involved orthogonal equivariance into constrained systems, which is nontrivial. Particularly, unifying equivariance is important and useful for practical applications across several domains, given that our experiments presented on MD17 and MoCap proof this. To address your concerns, the related discussion and comparison have been added in Section 2 in the revised paper.
> > >
> > > Regarding our extension of EGNN, we still believe that it is insightful and valuable. We have theoretically proved when and why our general form beyond EGNN can universally approximate any equivariant function. Our experiments also support the superiority of our method compared to EGNN.
> > >
> > >
> > > Once again, thank you very much for your further comments.
> > >
> > > Best，
> > >
> > > The authors

---

> ### Author Response · Authors · 2021-11-17
> **Welcome to comment**
>
> Dear reviewer,
>
> We have clarified the novelty of our paper. The justifications of both the experimental and theoretical contributions are also provided. Please feel free to ask any further questions.
>
> Thanks.

---

### Official Review · Reviewer_4PY9 · 2021-11-02

**Correctness:** 4
**Technical Novelty And Significance:** 2
**Empirical Novelty And Significance:** 3
**Recommendation:** 8
**Confidence:** 3

**Main Review:**

# Strenghts

- The introduction does a good job of motivating the need for the proposed work.
- The description of EGNN and the preliminary material more generally provide a good understanding of the relevant prior work.
- The model essentially relies on an object-centric view and the dynamics of each object are treated unitarily. This could also be thought of as a novel instance of object-centric-based learning.
- The theoretical analysis around the newly proposed equivariant layer is nice and useful.
- Since the EGNN layer is a particular case of the proposed one, the derived result also has theoretical implications for this prior work and improves our understanding of its limitations.
- The evaluation equips all the baselines with minimal information about the presence of the rigid objects (i.e. the edge-type indicator functions). This ensures a certain level of fairness of the evaluation methodology.
- I like that the evaluation is nicely factorized into different settings (e.g. small vs large training sets and results across different types of systems). This provides further insights into how the model performs in different regimes.
- Table 2 provides a nice empirical confirmation that the model works as expected and that the constraints are satisfied.
- The generalization of the proposed models to novel system configurations is tested in Table 3 and the results look good.
- The ablation study confirms the intuition / theoretical motivations provided for various design decisions described in earlier parts of the paper.

# Weaknesses

-  The evaluation only looks at a single simulated dataset. While these simulations are nice and insightful, I would have expected some real-world evaluation as well. Even the original work of Kipf considered a few real-world datasets. For instance, the authors could have looked at some motion capture data where different parts of the human body could be treated as rigid-body constraints.
- Related to the point above, the forward kinematics part of the model relies on having a perfect understanding of the nature of the constraint and relies completely on domain knowledge. This raises two issues: 1) What happens when the constraint is so complicated we might not be able to derive some forward kinematics equations like for the simple hinges and sticks? 2) This additional domain knowledge provides very important information that the baselines do not necessarily have. For these two reasons, it would have been useful to also consider a model where Equation (9) is learned by an abstract learnable function. This would have provided a model less reliant on very specific domain knowledge and more centered on the inductive bias of object-centric modeling.
- The considered constraints are relatively simple (i.e. small rigid bodies) and do not really test how the model can cope with more complex constraints. For instance, the sum in Equation (7) might increase too much in magnitude as the object is formed of many more parts. This naturally brings the question of whether the model can cope in practice with very large rigid objects?
- None of the reported results have standard errors around them. The experiments should be run at least across multiple initializations in order to assess the statistical significance of the results.



**Summary Of The Paper:**

The paper proposes a new graph neural network model for predicting the dynamics of n-body systems. The two main innovations of this model are:
1. Ability to handle systems with rigid-body constraints.
2. A new 3D transformation equivariant layer with universality properties.

**Summary Of The Review:**

Balancing the weaknesses and strengths from the list above, the paper is, in my view, at the threshold between acceptance and rejection. Since there is no neutral score I can select, I am voting for weak reject for now, but I am open to changing my score during the discussion period.

EDIT: Updated my score to 8.

---

> ### Author Response · Authors · 2021-11-14
> **Response to Reviewer 4PY9**
>
> We really appreciate the reviewer for listing the detailed strengths, which have clearly recognized the novelty and contributions of our paper. As requested by the reviewer, we have experimentally shown how to make our GMN generalizable to real and complex systems, including MD17 and Motion Capture. We acknowledge the suggestions by the reviewer, and have enhanced our paper accordingly.
>
> > **Q1: The evaluation in addition to a single simulated dataset.**
>
> The reviewer has mentioned the reference to the motion capture data, which really suits our goal and is able to verify the practicality of our method. Thank you for this nice suggestion! Besides implementing `CMU Motion Capture` similar to the work (Kipf et al.), we have also carried out extra evaluations on a real-world molecular dataset `MD17`. As reported in **G1**, we delightfully find that GMN is generally more effective against other equivariant models on these two tasks. These results can support the generalization ability of GMN to complex systems. The new results have been added into Section 4.2 in the paper.
>
> > **Q2: The implementation of the forward kinematics.**
>
> > Q2.1: What happens when the constraint is so complicated we might not be able to derive some forward kinematics equations like for the simple hinges and sticks?
>
> Please see **G2** for how GMN can be applied to complex systems without the need of deriving the exact kinematics.
>
> > Q2.2 The learnable Equation (9).  (``This part has been renewed in the next response. Please skip it if you have not read it.``)
>
> We are thankful to the reviewer for suggesting us to implement Eq. (9) (the FK equation) by an abstract learnable function to reduce the reliance on domain knowledge. This indeed introduces an interesting point on how to balance the usage of domain knowledge (or the so-called inductive bias). While this can be discussed in a wider context, here restricted to our problem, we would like to highlight that involving the domain knowledge of the rigid constraint into GMN is still what we pursue in the first place, and it requires fewer training samples and ensures more data efficiency compared to the data-driven counterparts. Specifically, as we explained in **G2**, using the current hand-crafted form of Eq. (9) is still capable of handling complex systems. Of course, we are interested in exploring learnable FK in the future work.
>
> > **Q3: The considered constraints are relatively simple.**
>
> Thanks for the comment. As explained above in **G1** and **G2**, upon the simple constraints by stick modeling, our GMN is able to cope with more complex systems by kinematics decomposition, without the need of modifying Eq. (7). This will avoid the issue of the increase of the magnitude for the sum in Eq. (7), as mentioned by the reviewer.
>
> > **Q4: Statistical significance of the results.**
>
> Thanks for the kind reminder. We have added the standard deviations for all methods over 3 random runs in Table 1. It shows that the standard deviation of GMN is small, validating the statistical significance of its improvement over other methods.

---

> > ### Author Response · Authors · 2021-11-15
> > **The experiments on learnable Equation (9) have been added**
> >
> > In Q2, the reviewer has suggested us to implement Eq. (9) (the FK equation) by an abstract learnable function to reduce the reliance on domain knowledge.  After we submitted the previous response, we have reconsidered this question again. It is indeed instrumental to discuss whether a learnable black-box function, which requires less domain knowledge, could also yield competitive performance, and if our hand-crafted FK still shows advantage over the learnable counterpart. To answer these questions, we replace the hand-crafted part (Eq. (7-9) as well as the Euler angle computations in Sec. 3.2) with the following equations:
> > $v_i^l = \phi (h_i^{l-1})v_i^{l-1}$ +$ \rho (q'', x_{ki}^{l-1}, f_i^l),x_i^l = x_i^{l-1} + v_i^l$
> > where $q''=\ddot{q}_{k}^l$, $\rho$ is the equivariant message passing layer we propose in Sec. 3.3. By this design, the parmeterized FK preserves its equivariant property (and the theoretical universality), and compared to EGNN, it additionally leverages the information from the object-level generalized coordinates $\ddot{q}_k^l$. We denote this variant of GMN as GMN-L. Moreover, the parameterized FK inevitably loses the constraint-preserving property compared with the exact FK, therefore we also augment it with explicit constraint regularization, akin to what we did to EGNNReg. We hence denote this variant as GMN-LReg.
> >
> > We evaluate the performance of GMN-L, GMN-LReg and compare them with GMN with exact FK as well as EGNN and EGNNReg in the table below. We have contained the results of the requested black-box baseline in Table 3 in the main body of the revised paper.
> >
> >
> >
> > | \|Train\| = 500 | (1,2,0)  | (2,0,1)  | (3,2,1)  | (0,10,0) | (5,3,3)  |
> > | --------------- | -------- | -------- | -------- | -------- | -------- |
> > | EGNN            | 2.81     | 2.27     | 4.67     | 4.75     | 4.59     |
> > | EGNNReg         | 2.94     | 2.66     | 7.01     | 5.03     | 6.31     |
> > | GMN-L           | **2.32** | **2.09** | **3.19** | 3.88     | 4.34     |
> > | GMN-LReg        | 2.52     | 2.23     | 3.34     | **3.67** | **4.31** |
> > | GMN             | **1.84** | **2.02** | **2.48** | **2.92** | **4.08** |
> >
> >
> >
> > | \|Train\| = 1500 | (1,2,0)  | (2,0,1)  | (3,2,1)  | (0,10,0) | (5,3,3)  |
> > | ---------------- | -------- | -------- | -------- | -------- | -------- |
> > | EGNN             | 2.59     | 1.86     | 2.54     | 2.79     | 3.25     |
> > | EGNNReg          | 2.74     | 1.58     | 2.62     | 3.03     | 3.07     |
> > | GMN-L            | 1.93     | **1.56** | **2.28** | 2.72     | 3.03     |
> > | GMN-LReg         | **1.91** | 1.88     | 2.49     | **2.61** | **3.00** |
> > | GMN              | **1.68** | **1.47** | **2.10** | **2.32** | **2.86** |
> >
> > We interestingly find that GMN-L consistently outperforms EGNN in various settings (as well as the regularized version), which again verifies both the validity of our proposed equivariant message passing layer and the efficacy of leveraging object-level message (i.e., $\ddot{q}_k^l$) for the inference of FK. At the same time, GMN-L and GMN-LReg yield a minor gap with GMN, showing the evidence that hardcoding the constraints replaces a nontrivial amount of learning complexity. Even so, GMN-L  ``would have provided a model less reliant on very specific domain knowledge and more centered on the inductive bias of object-centric modeling``, as suggested by the reviewer.
> >
> > We sincerely thank the reviewer again for motivating us to design the learnable FK, which better strengthens our contribution of the equivariant message-passing layer.

---

> > > ### Comment · Reviewer_4PY9 · 2021-11-18
> > > **Response to Authors**
> > >
> > > Thank you for your detailed response and additions to the paper. I think the paper is in much better shape now and the effort the authors have put into addressing the comments is commendable. Therefore, I will raise my score to 6.
> > >
> > > A couple of issues prevent me from assigning a higher score:
> > > - The black box learning of the forward kinematics right now seems to have a rather peripheric role in the paper. This is somewhat expected given that the paper did not have it at the beginning of the review period, but it remains something not fully explored and hard to address completely during the discussion period. Most importantly, the black-box model is evaluated only in the synthetic synthetic setting, but not in real-world examples.
> > > - The way the stick constraints were added to the real-world experiments as partial constraints is a very nice trick. However, this also feels like it could have been much deeply explored. What happens if you decompose into slightly bigger objects rather than just sticks? And related to the above, what happens if black-box kinematics are used?

---

> > > > ### Author Response · Authors · 2021-11-19
> > > > **Further Responses to Reviewer 4PY9 (Page 2)**
> > > >
> > > >
> > > > > ``The way the stick constraints were added to the real-world experiments as partial constraints is a very nice trick. However, this also feels like it could have been much deeply explored. What happens if you decompose into slightly bigger objects rather than just sticks? And related to the above, what happens if black-box kinematics are used?``
> > > >
> > > > Thank you for this great question. It is indeed possible to decompose into bigger objects rather than just sticks. As a comparison, we further adopt the hinge-wise decomposition (*i.e.*, decompose the system into particles and hinges), and investigate the performance on both MD17 and Motion Capture. We denote this model as **GMN-LH**, where H stands for hinges.
> > > >
> > > >
> > > > The results on **MD17**:
> > > >
> > > >
> > > > |        | Aspirin           | Benzene            | Ethanol           | Malonaldehyde      | Naphthalene       | Salicylic         | Toluene            | Uracil            |
> > > > | ------ | ----------------- | ------------------ | ----------------- | ------------------ | ----------------- | ----------------- | ------------------ | ----------------- |
> > > > | EGNN   | 14.41$\pm$0.15    | 62.40$\pm$0.53     | 4.64$\pm$​​0.01     | 13.64$\pm$0.01     | 0.47$\pm$0.02     | 1.02$\pm$0.02     | 11.78$\pm$0.07     | 0.64$\pm$0.01     |
> > > > | GMN    | 10.14$\pm$​​0.03    | **48.12**$\pm$​0.40 | 4.83$\pm$0.01     | 13.11$\pm$​​0.03     | **0.40**$\pm$​​0.01 | 0.91$\pm$​​0.01     | **10.22**$\pm$​0.08 | **0.59**$\pm$0.01 |
> > > > | GMN-L  | **9.76**$\pm$​0.11 | 54.17$\pm$​0.69     | 4.63$\pm$​0.01     | **12.82**$\pm$​0.03 | 0.41$\pm$​0.01     | **0.88**$\pm$0.01 | 10.45$\pm$​​0.04     | **0.59**$\pm$0.01 |
> > > > | GMN-LH | 10.25$\pm$0.06    | 52.02$\pm$0.97     | **4.62**$\pm$​0.01 | 12.83$\pm$​0.03     | 0.41$\pm$0.01     | 1.03$\pm$0.01     | 10.81$\pm$0.14     | **0.59**$\pm$0.01 |
> > > >
> > > >
> > > >
> > > > The results on **Motion Capture**:
> > > >
> > > >
> > > >
> > > > | EGNN         | GMN              | GMN-L        | GMN-L H          |
> > > > | ------------ | ---------------- | ------------ | ---------------- |
> > > > | 59.1$\pm$2.1 | **43.9**$\pm$1.1 | 50.9$\pm$0.7 | **48.7**$\pm$​1.1 |
> > > >
> > > >
> > > >
> > > > On MD17, GMN-LH yields a little bit worse performance than GMN-L on several molecules, while giving desirable results on Ethanol and Benzene. On Motion Capture, GMN-LH outperforms GMN-L by a small gap, while is still worse than GMN.
> > > > By default, we still encourage to perform the decomposition via sticks as sticks are actually the basic building blocks of hinges and other larger rigid objects.
> > > > We believe that it is indeed interesting to investigate how to "optimally" decompose a given complex graph. However, we had better leave this for future exploration, as our goal is mainly on involving the object-level message passing into EGNN and our experiments above do support this benefit by either the hand-crafted or the black-box strategy.
> > > >
> > > > We are very willing to further clarify if there is still any concern unresolved.

---

> > > > > ### Comment · Reviewer_Dcng · 2021-11-21
> > > > > **Thank you**
> > > > >
> > > > > Thank you very much, barring any other concerns raised in discussion with other reviewers I'm open to increasing my score to a 7 now.

---

> > > > > > ### Author Response · Authors · 2021-11-21
> > > > > > **Thank you very much!**
> > > > > >
> > > > > > Thank you! Your suggestions help improve our work a lot, and we really enjoy the discussion with you.
> > > > > >
> > > > > > Thanks again!

---

> > > > > ### Comment · Reviewer_4PY9 · 2021-11-21
> > > > > **Response to authors**
> > > > >
> > > > > Thank you for addressing these last two points! While some concerns about novelty reported by @Reviewer F8Jp remain, I think the paper is a mix of interesting ideas, and, taken together, they could be a valuable contribution to the conference. Now that the fully-learned version of the model has been explored in more/sufficient detail, I will further raise my score to Accept (8).

---

> > > > > > ### Author Response · Authors · 2021-11-21
> > > > > > **Thank you very much!**
> > > > > >
> > > > > > Thank you! Our paper won’t be better without the nice suggestions by you! Really appreciate your constructive comments.
> > > > > >
> > > > > > Thanks again!

---

> > > > ### Author Response · Authors · 2021-11-19
> > > > **Further Responses to Reviewer 4PY9 (Page 1)**
> > > >
> > > > Thank you for your supportive feedback! We further add more experiments in order to address your concerns for a higher score.
> > > >
> > > > >  ``The black box learning of the forward kinematics right now seems to have a rather peripheric role in the paper. This is somewhat expected given that the paper did not have it at the beginning of the review period, but it remains something not fully explored and hard to address completely during the discussion period. Most importantly, the black-box model is evaluated only in the synthetic setting, but not in real-world examples.``
> > > >
> > > >
> > > > Thanks for your constructive comment! According to your suggestion, we further evaluate the black-box forward kinematics model (GMN-L) on the two real-world datasets MD17 and Motion Capture. We decompose the molecules and motion graphs into particles and sticks following the same kinematics decomposition trick as **G2**. Similar to the implementation for the synthetic setting, GMN-L shares the same common function $\phi_{1}$ with EGNN, and the functions $\phi_{1}$ and $\phi_2$ with GML, for exactly fair comparisons in all experiments we conducted.
> > > >
> > > >
> > > > The results on **MD17**:
> > > >
> > > >
> > > >
> > > > |         | Aspirin           | Benzene            | Ethanol           | Malonaldehyde      | Naphthalene       | Salicylic         | Toluene            | Uracil            |
> > > > | ------- | ----------------- | ------------------ | ----------------- | ------------------ | ----------------- | ----------------- | ------------------ | ----------------- |
> > > > | RF      | 10.94$\pm$0.01    | 103.72$\pm$1.29    | 4.64$\pm$​​0.01     | 13.93$\pm$0.03     | 0.50$\pm$0.01     | 1.23$\pm$0.01     | 10.93$\pm$0.04     | 0.64$\pm$0.01     |
> > > > | TFN     | 12.37$\pm$0.18    | 58.48$\pm$1.98     | 4.81$\pm$0.04     | 13.62$\pm$0.08     | 0.49$\pm$0.01     | 1.03$\pm$0.02     | 10.89$\pm$0.01     | 0.84$\pm$0.02     |
> > > > | SE3-Tr. | 11.12$\pm$0.06    | 68.11$\pm$0.67     | 4.74$\pm$0.13     | 13.89$\pm$0.02     | 0.52$\pm$0.01     | 1.13$\pm$0.02     | 10.88$\pm$0.06     | 0.79$\pm$0.02     |
> > > > | EGNN    | 14.41$\pm$0.15    | 62.40$\pm$0.53     | 4.64$\pm$​​0.01     | 13.64$\pm$0.01     | 0.47$\pm$0.02     | 1.02$\pm$0.02     | 11.78$\pm$0.07     | 0.64$\pm$0.01     |
> > > > | EGNNReg | 13.82$\pm$0.19    | 61.68$\pm$0.37     | 6.06$\pm$0.01     | 13.49$\pm$0.06     | 0.63$\pm$0.01     | 1.68$\pm$0.01     | 11.05$\pm$0.01     | 0.66$\pm$0.01     |
> > > > | GMN     | 10.14$\pm$​​0.03    | **48.12**$\pm$​0.40 | 4.83$\pm$0.01     | 13.11$\pm$​​0.03     | **0.40**$\pm$​​0.01 | 0.91$\pm$​​0.01     | **10.22**$\pm$​0.08 | **0.59**$\pm$0.01 |
> > > > | GMN-L   | **9.76**$\pm$​0.11 | 54.17$\pm$​0.69     | **4.63**$\pm$0.01 | **12.82**$\pm$​0.03 | 0.41$\pm$​0.01     | **0.88**$\pm$0.01 | 10.45$\pm$​​0.04     | **0.59**$\pm$0.01 |
> > > >
> > > >
> > > >
> > > >
> > > > The results on **Motion Capture**:
> > > >
> > > > | GNN          | TFN          | SE3-Tr.      | RF            | EGNN         | EGNNReg      | GMN              | GMN-L            |
> > > > | ------------ | ------------ | ------------ | ------------- | ------------ | ------------ | ---------------- | ---------------- |
> > > > | 67.3$\pm$​1.1 | 67.3$\pm$1.1 | 60.9$\pm$0.9 | 197.0$\pm$1.0 | 59.1$\pm$2.1 | 59.5$\pm$2.2 | **43.9**$\pm$1.1 | **50.9**$\pm$0.7 |
> > > >
> > > >
> > > > Surprisingly, GMN-L showcases very competitive performance on both real-world datasets. On MD17, GMN-L surpasses GMN (with hand-crafted FK) on 4 of the 8 molecules, and it remarkably outperforms EGNN in all cases except Ethanol. Even for Ethanol, GMN-L still yields slightly better performance than EGNN, while GMN is worse. As for the Motion Capture dataset, GMN-L outperforms other equivariant baselines by a large margin, although it is inferior to GMN. These results are reasonable, as the learnability of constraints by GMN-L can better fit the case when the length constraint is NOT strictly required. For example in the molecule Ethanol, the vibration of the bond length will somehow influence the position prediction since Ethanol's molecular structure is relatively small. On the other hand, when hard constraints prevail, for example on the motion capture task, GMN-L will become worse than GMN, probably because the domain knowledge of the hand-crafted FK plays an important role in this case.
> > > >
> > > > Overall, the results here are quite exciting, and both GMN and GMN-L are proven effective on real-world data. We thank the reviewer again for motivating us to explore more on the black-box learning of the forward kinematics. We have added these insightful discussions in Section 4.2 in the revised paper. We also contain the learnable FK as a supplement to the current implementation of hand-crafted FK in Section 3.4.

---

> ### Author Response · Authors · 2021-11-17
> **Welcome to comment**
>
> Dear reviewer,
>
> We have provided responses regarding the experiments on two real-world applications. Besides, we have conducted evaluations on learnable FK. Please feel free to ask any questions.
>
> Thanks.

---

### Author Response · Authors · 2021-11-14
**General responses (Page 1)**

We sincerely thank all reviewers. We have revised the paper to reflect all reviewers’ suggestions, since the revisions are allowed during the rebuttal period according to the ICLR offical guidance.


## G1: Additional evaluations on two real-world datasets.

The reviewers have raised their questions on if our method can be generalized to more complex systems besides sticks and hinges. We thank the reviewers for raising this concern. Our GMN can indeed fulfill this extension. To show this, we additionally evaluate GMN on two real-world datasets: `MD17` (Chmiela et al., 2017) and `CMU Motion Capture` (CMU, 2003). The new experiments have been added in Section 4.2.

### Results on `MD17`
`MD17` involves the atom-wise 3D trajectories of eight small organic molecules (such as Benzene, Aspirin, etc) generated via molecular dynamics simulation. `MD17` is previously used for property prediction task. Our goal is predicting the future positions of the atoms starting from their initial positions for each molecule. By this evaluation, we are able to demonstrate the potential application of GMN on molecular dynamics prediction which is a vital task in various applications, such as drug discovery. Each molecule can be regarded as a complex system consisting of atoms and bonds. According to Chemistry, the bond length usually changes slightly and can be considered as being fixed (we have actually computed the mean square change of the bond length along each trajectory and found its small ranging from $10^{-3}$ to $10^{-4}$). It thus motivates us to realize the bond constraint by our stick model. In particular, we describe each molecule with a set of particles and sticks by the kinematics decomposition trick (the details are provided in **G2**), and then employ GMN for the dynamics modeling. The full details are provided in Appendix E. The results are recorded in the following table.



|         | Aspirin            | Benzene            | Ethanol           | Malonaldehyde      | Naphthalene       | Salicylic         | Toluene            | Uracil            |
| ------- | ------------------ | ------------------ | ----------------- | ------------------ | ----------------- | ----------------- | ------------------ | ----------------- |
| RF      | 10.94$\pm$0.01     | 103.72$\pm$1.29    | **4.64**$\pm$​0.01 | 13.93$\pm$0.03     | 0.50$\pm$0.01     | 1.23$\pm$0.01     | 10.93$\pm$0.04     | 0.64$\pm$0.01     |
| TFN     | 12.37$\pm$0.18     | 58.48$\pm$1.98     | 4.81$\pm$0.04     | 13.62$\pm$0.08     | 0.49$\pm$0.01     | 1.03$\pm$0.02     | 10.89$\pm$0.01     | 0.84$\pm$0.02     |
| SE3-Tr. | 11.12$\pm$0.06     | 68.11$\pm$0.67     | 4.74$\pm$0.13     | 13.89$\pm$0.02     | 0.52$\pm$0.01     | 1.13$\pm$0.02     | 10.88$\pm$0.06     | 0.79$\pm$0.02     |
| EGNN    | 14.41$\pm$0.15     | 62.40$\pm$0.53     | **4.64**$\pm$​0.01 | 13.64$\pm$0.01     | 0.47$\pm$0.02     | 1.02$\pm$0.02     | 11.78$\pm$0.07     | 0.64$\pm$0.01     |
| EGNNReg | 13.82$\pm$0.19     | 61.68$\pm$0.37     | 6.06$\pm$0.01     | 13.49$\pm$0.06     | 0.63$\pm$0.01     | 1.68$\pm$0.01     | 11.05$\pm$0.01     | 0.66$\pm$0.01     |
| GMN     | **10.14**$\pm$​0.03 | **48.12**$\pm$​0.40 | 4.83$\pm$0.01     | **13.11**$\pm$​0.03 | **0.40**$\pm$​​0.01 | **0.91**$\pm$​0.01 | **10.22**$\pm$​0.08 | **0.59**$\pm$0.01 |


GMN outperforms other competitive equivariant models on 7 of the 8 molecules, showing its potential for modeling real-world systems. Particularly, on molecules with complex structures (*e.g.*, Aspirin, Benzene, and Salicylic), the improvement of GMN against the other methods is more significant, showcasing the benefit of constraint modeling on the bonds. Yet, we also observe that models involving constraints (GMN and EGNNReg) perform worse than others on Ethanol, possibly because Ethanol is a relatively small and simple molecule (it only contains 2 bonds), where considering bond-length constraint makes less benefit but somehow restrains the training.

### Results on `CMU Motion Capture`
Besides `MD17`, we have also evaluated the performance of GMN on `CMU Motion Capture` data where different parts of the human body could be treated as hard rigid-body constraints. The task is to predict the future states of the human joints given their initial states.
After applying the kinematics decomposition trick (see **G2**), each motion graph is decomposed as a set of particles and sticks, which is fed to GMN for prediction. We report the results in the following table (or Table 5 in the revised paper), where GMN is clearly superior to other compared methods. Please see the details in Section 4.2 in the modified version.



| GNN          | TFN          | SE3-Tr.      | RF            | EGNN         | EGNNReg      | GMN              |
| ------------ | ------------ | ------------ | ------------- | ------------ | ------------ | ---------------- |
| 67.3$\pm$​1.1 | 67.3$\pm$1.1 | 60.9$\pm$0.9 | 197.0$\pm$1.0 | 59.1$\pm$2.1 | 59.5$\pm$2.2 | **43.9**$\pm$1.1 |

---

> ### Author Response · Authors · 2021-11-14
> **General responses (Page 2)**
>
> ## G2. The implement of the forward kinematics.
>
> It is true that we manually build the forward kinematics Eq. (9) of the stick (and hinge), which relies on the domain knowledge of the underlying physics. However, for complex systems, we no longer require to derive the kinematics of the entire system. Instead, we propose kinematics decomposition, a simple yet effective trick that can decompose each input system (an arbitrary graph) into particles and sticks. Taking the `MD17` dataset as an example, for each molecule, we select certain bonds as sticks and the remaining atoms as isolated particles; in this way, we obtain a set of particles and sticks. Note that different sticks are not allowed to intersect; otherwise, it will generate two values for the intersecting particle of two sticks and cause ambiguity if these two values are distinct. Although this kind of kinematics decomposition will only maintain partial constraints, it greatly enlarges the application scope of our current formulation Eq. (5-9) without any revision. More importantly, our experiments reported in **G1** do verify that GMN by this formulation is sufficient to surpass other methods on complex systems like molecules. On `CMU Motion Capture`, since the motion graph contains no circle and is of the tree-like structure, it is tractable to derive the exact forward kinematics by recursive kinematics computation from the root node. Yet, we still encourage the usage of the above kinematics decomposition for its easy implementation and compatibility with our GMN.

---

### Decision · Program_Chairs · 2022-01-20

**Decision:**

Accept (Poster)

**Comment:**

The manuscript develops a new kind of graph neural network (a Graph Mechanics Network; GMN) that is particularly well suited to representing and making predictions about physical mechanics systems (and data with similar structure). It does so by developing a way to build geometric constraints implicitly and naturally into the forward kinematics of the network, while still allowing for effective learning from data. The manuscript proves some essential properties of the new architecture and runs experiments both with simulated particles, hinges, sticks (and their combination), as well as with motion capture data.
Reviewers were generally impressed by the writing and clarity of the work, as well as the main results. In addition, in those cases where reviewers thought that the experiments were lacking, the authors delivered effective new experiments to address those concerns (e.g. looking  at mocap and molecular datasets). One reviewer initially scores the manuscript as a Reject/3 on the basis of concerns about novelty and the scope of the theoretical and experimental contributions of the paper. However, they adjust their score 3->5 based on the rebuttal presented by the authors (including new experiments). The reviewer also downgrades their certainty (from 3->2) on the basis of the engagement from reviewers offering higher scores.
Overall, the manuscript presents a promising contribution to the graph networks literature and I agree with the general consensus in favour of publication.